

# Dynamically downscaled seasonal ocean forecasts for North American East Coast ecosystems

Andrew C. Ross[1], Charles A. Stock[1], Vimal Koul[2], Thomas L. Delworth[1], Feiyu Lu[1], Andrew Wittenberg[1], and Michael A. Alexander[3]

[1]NOAA/OAR/Geophysical Fluid Dynamics Laboratory, 201 Forrestal Road, Princeton, NJ 08540, USA
[2]Princeton University Cooperative Institute for Modeling the Earth System, 201 Forrestal Road, Princeton, NJ 08540, USA
[3]NOAA/OAR/Physical Sciences Laboratory, 325 Broadway, Boulder, CO 80305, USA

**Correspondence:** Andrew C. Ross (andrew.c.ross@noaa.gov)

**Abstract.** Using a 1/12° regional model of the Northwest Atlantic Ocean (MOM6-NWA12), we downscale an ensemble of retrospective seasonal forecasts from a 1° global forecast model. To evaluate whether downscaling improved the forecast skill for surface temperature and salinity and bottom temperature, the global and downscaled forecasts are compared with each other and with a reference forecast of persistence using anomaly correlation. Both sets of forecasts are also evaluated on the basis

of mean bias and ensemble spread. We find that downscaling significantly improved the forecast skill for monthly sea surface temperature anomalies in the Northeast U.S. Large Marine Ecosystem, a region that global models have historically struggled to predict skillfully. The downscaled SST predictions for this region were also more skillful than the persistence baseline across most initialization months and lead times. Although some of the SST prediction skill in this region stems from the recent, rapid warming trend, prediction skill above persistence is generally maintained after removing the contribution of the trend,

and patterns of skill suggestive of predictable processes are also preserved. While downscaling mainly improved SST skill anomaly prediction skill in the Northeast U.S. region, it improved bottom temperature and sea surface salinity anomaly skill across many of the marine ecosystems along the North American East Coast. Although improvements in anomaly prediction via downscaling were ubiquitous, the effects of downscaling on prediction bias were mixed. Downscaling generally reduced the mean surface salinity biases found in the global model, particularly in regions with sharp salinity gradients (the Northern

Gulf of Mexico and the Northeast U.S.). In some cases, however, downscaling amplified the surface and bottom temperature biases found in the global predictions. We discuss several processes that are better resolved in the regional model and contribute to the improved skill, including the autumn reemergence of temperature anomalies and advection of water masses by coastal currents. Overall, the results show that a downscaled, high resolution model can produce improved seasonal forecast skill by representing fine-scale processes that drive predictability.

**1 Introduction**

The development of seasonal climate forecasting systems, which provide information about predicted climate conditions on timescales ranging from months to years, has sparked new applications to anticipate the response of marine ecosystems to predictable climate variability and provide information to inform the management of living marine resources (Jacox et al.,



2020; Payne et al., 2017; Tommasi et al., 2017). Forecasts of sea surface temperatures from these seasonal forecast systems,
which are based on global climate models with coupled atmosphere and ocean components, are skillful across many large
marine ecosystems (LMEs) (Hervieux et al., 2017; Stock et al., 2015). In many regions, global models are also capable of
predicting the occurrence and intensity of marine heatwaves (Jacox et al., 2022) and chlorophyll anomalies at the ocean surface
(Park et al., 2019). A variety of processes contribute to predictability in different regions, including the progression of coastal
waves and the reemergence of prior anomalies (Jacox et al., 2020).

The skill of seasonal predictions for LMEs along the North American East Coast (NAEC), including the Northeast U.S.
and Scotian Shelf LMEs, is typically much lower than many other regions in global forecast models (e.g., Stock et al., 2015).
McAdam et al. (2022) suggested that the lack of skill may be due to coarse model resolution resulting in poor initialization
or inability to maintain sharp fronts in the NAEC and other western boundary current regions. For the NAEC in particular,
it is well known that models with resolution coarser than 1/10° are unable to accurately simulate the Gulf Stream and its
separation from the continental shelf, typically resulting in severe temperature biases north and west of the current along the
shelf and coast (Chassignet and Garaffo, 2001; Chassignet and Xu, 2017; Saba et al., 2016). Furthermore, compared to many
other regions like the North American West Coast, predictability for the East Coast is hindered by its weaker teleconnections
to ENSO and other predictable climate modes at seasonal time scales. The seasonal variability of ocean temperature anomalies
in the North Atlantic basin is strongly connected to the North Atlantic Oscillation (NAO), but this mode is only moderately
predictable at seasonal time scales (with dynamical model skill generally requiring large ensembles; Smith et al., 2020), and
the contribution of NAO to ocean temperature predictability involves features and processes that are often poorly simulated in
coarse resolution models, including modulation of the Gulf Stream (Sanchez-Franks et al., 2016; Shin and Newman, 2021) and
reemergence of anomalies from the previous winter sequestered below the mixed layer (Sukhonos and Alexander, 2023).

To the southwest, along the coast of the Gulf of Mexico, the predictability of ocean conditions is potentially enhanced
by variability connected to the El Niño-Southern Oscillation (e.g., Alexander and Scott, 2002; Gomez et al., 2019). Ideal-
ized experiments have also shown the potential for 1–3 months of predictability of the Loop Current in the Gulf of Mexico
(Dukhovskoy et al., 2023; Haley et al., 2023). High resolution downscaled forecasts are likely necessary to realize this potential
predictability; horizontal resolution of at least 10–25 km is necessary to simulate Loop Current eddies, for example (Oey et al.,
2013).

Seasonal forecast models based on empirical relationships with observed temperature have shown some temperature predic-
tion skill for the NAEC. Shin and Newman (2021) developed a linear inverse model to predict SST and found that the model
was significantly more skillful than a multimodel mean of global dynamical forecast models at predicting SST anomalies in the
Northeast U.S. and Scotian Shelf LMEs, particularly at longer lead times. Chen et al. (2021) found that bottom temperatures
in NA coastal regions could be skillfully predicted by combining persistence of recent conditions with empirically determined
advection from nearby regions. However, both of these models are univariate (they predict only surface or bottom temperature),
are constrained by the availability and accuracy of historical observations, and may become unreliable if ocean currents, mixed
layer depths, and other properties that influence the seasonal variability of ocean temperatures change in the future.



Dynamical downscaling of global model predictions with a high resolution regional model could provide accurate seasonal predictions while avoiding many of the drawbacks of statistical/empirical predictions. In a pioneering application, Siedlecki et al. (2016) developed skillful dynamically downscaled forecasts of ocean temperature, oxygen, and acidification for the Northern California Current System. For the NAEC, dynamical downscaling has been used to produce short-term forecasts for the next few days (Wilkin et al., 2018) and long-term projections of the effects of climate change (Alexander et al., 2020; Han et al., 2019; Rutherford et al., 2023), but not to produce seasonal forecasts.

Dynamical downscaling does not guarantee increased forecast skill, however. Dynamically downscaled simulations often resemble the solution of the coarse model they are downscaling, which brings into question how much value they add (e.g., Ghantous et al., 2020). Although dynamical downscaling offers the possibility of running multiple ensemble members to provide information about forecast uncertainty, the substantial computational cost of high resolution models used for downscaling has constrained the ensemble sizes of past climate downscaling efforts (Drenkard et al., 2021). In an analysis of seasonal forecasts for the California Current System, Brodie et al. (2023) found that although a small ensemble of downscaled forecasts outperformed a subset of the same size from a much larger ensemble of coarse resolution forecasts, the full ensemble of coarse resolution forecasts outperformed the small downscaled ensemble. Furthermore, in some cases high ocean resolution has been found to actually degrade forecast skill (Sandery and Sakov, 2017; Thoppil et al., 2021).

In this study, we test the hypothesis that a high resolution dynamical ocean model will improve the skill of seasonal forecasts for the NAEC relative to a lower resolution global model. To do this, we dynamically downscale 29 years of retrospective forecast simulations produced by GFDL's Seamless System for Prediction and EArth System Research (SPEAR) model, which uses a coupled ocean component with 1° resolution. We downscale the SPEAR seasonal forecasts with a 1/12° model of the Northwest Atlantic Ocean built with the regional modeling capabilities of the MOM6 ocean model (MOM6-NWA12), which has recently been assessed for accuracy at simulating ocean temperatures, the position and variability of the Gulf Stream, and other relevant ocean features along the NAEC in a reanalysis-forced historical simulation (Ross et al., 2023). To the best of our knowledge, this is the first attempt to dynamically downscale seasonal ocean forecasts for the Northwest Atlantic Ocean.

In the sections that follow, we describe the methods used to downscale the SPEAR forecasts and evaluate the retrospective prediction skill (Section 2). The presentation of the model results (Section 3) focuses on seasonal prediction skill for surface and bottom temperature and surface salinity in the downscaled simulations relative to the global simulations. In the discussion (Section 4), we examine drivers of predictability and skill and discuss potential future improvements to the forecasting system. We conclude by noting the promising potential for these seasonal forecasts to improve the management of living marine resources along the NAEC.

## 2 Methods

### 2.1 Numerical models

The high-resolution regional model used in this study is a 1/12° MOM6-based model of the Northwest Atlantic Ocean (MOM6-NWA12; Ross et al., 2023). Briefly, MOM6-NWA12 is a regional ocean-sea ice model with explicit tides that spans a wide





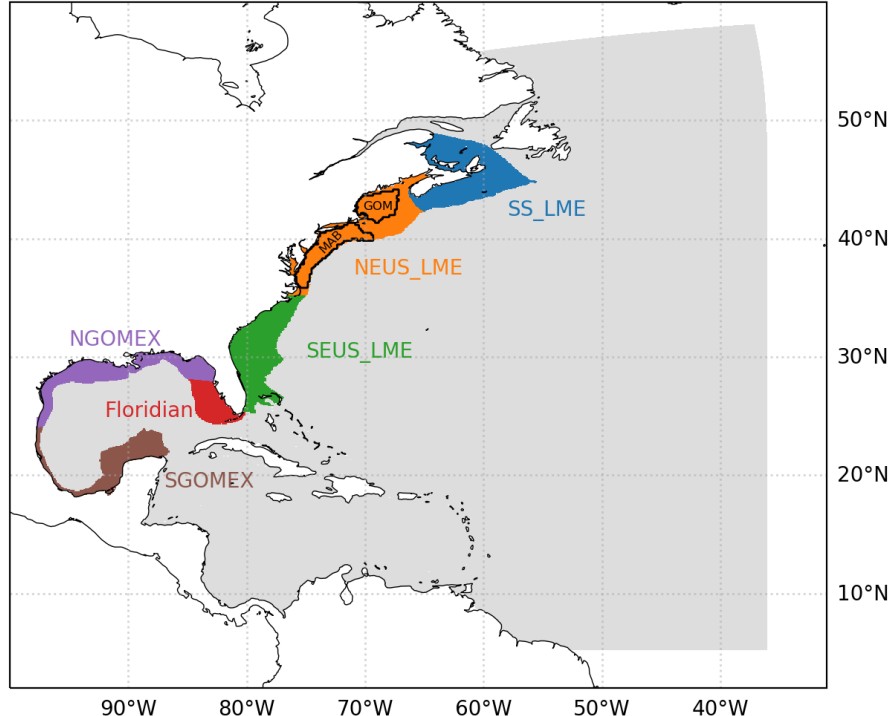

**Figure 1.** Map of the NWA12 ocean model domain (light gray shading) and the six marine ecosystems for which spatial averages were computed in the results: the Scotian Shelf (SS), Northeast U.S. (NEUS), and Southeast U.S. (SEUS) Large Marine Ecosystems (LMEs), and the Floridian and Northern and Southern Gulf of Mexico (NGOMEX, SGOMEX) ecoregions. Black outlines within the NEUS LME show two Ecological Production Units: the Mid-Atlantic Bight (MAB) and the Gulf of Maine (GOM).

domain including the Caribbean Sea, Gulf of Mexico, and North American East Coast (Figure 1). The model configuration that we use here is essentially the same as the one evaluated by comparing a reanalysis-forced historical simulation of the model with observations in Ross et al. (2023), including the same domain, vertical coordinates, parameterization choices, and parameter values. The only difference is that the version used here does not include coupled biogeochemistry. The model is

forced at the surface by atmospheric data, by river discharge from land, and at the ocean open boundaries. The wide extent of the model domain was designed to reduce the influence of the model being downscaled on the coastal and shelf environments and the western boundary current system by placing the ocean open boundaries far from these regions of interest. In the reanalysis-forced simulation spanning 1993 to 2019, Ross et al. (2023) found that the model was capable of accurately reproducing mean conditions of temperature, salinity, mixed layer depth, and other physical properties. The model also simulated many aspects

of historical variability, although some of the recent extreme warming trends in the region were underestimated.

The high-resolution regional model was used to downscale lower-resolution global retrospective forecasts produced by GFDL's SPEAR model (Delworth et al., 2020). As described in Lu et al. (2020), the SPEAR retrospective forecasts were run



using a coupled atmosphere-land and ocean-sea ice model with a nominal 0.5° resolution for the atmosphere and 1° resolution for the ocean (the SPEAR_MED configuration). The suite of retrospective forecasts from SPEAR starts in 1991 and extends
to the present, with recent and ongoing forecasts contributed to the North American Multi-Model Ensemble (NMME). Each SPEAR retrospective seasonal forecast was run for one year. Earlier forecasts contained 15 ensemble members, while more recent runs contributed to the NMME contain 30.

## 2.2 Downscaled retrospective forecast simulations

MOM6-NWA12 was initialized and used to run downscaled retrospective forecast simulations four times per year (beginning
on the first day of the month at the start of each meteorological season—March, June, September, and December) for every year from 1993 through 2021. Initial conditions for the forecasts were derived from an analysis simulation of the regional MOM6-NWA12 model forced by the same atmosphere (ERA5), ocean (GLORYS12), and river (GloFAS) reanalyses as in Ross et al. (2023). Additional details on the variables and boundary conditions used are provided in Ross et al. (2023). One difference is that in the analysis simulation, temperature and salinity throughout the model domain were nudged towards monthly means
from the GLORYS12 reanalysis with a 90 day damping time scale. Although the free-running simulation in Ross et al. (2023) successfully recreated many aspects of observed ocean variability and change, the addition of nudging helps maximize the accuracy of the initial conditions, while the moderate nudging time scale is intended to keep the simulation from being nudged too far away from the model's attractor and inducing a drift during the forecast when the nudging is removed. Instantaneous conditions saved at the beginning of each month of the historical simulation were used as initial conditions for the retrospective
forecast experiments. For each initialization date, 10 ensemble members were integrated forward for one year. Each member used atmospheric forcing from a different SPEAR forecast member (discussed next) but began with the same initial conditions from the analysis simulation. Using the same initial conditions for each ensemble member will reduce the ensemble spread; however, as we show in the results (Section 3), the ensemble members quickly diverge after initialization and the effect of the identical initial conditions is minimal. We acknowledge, though, that deriving the initial conditions from a data assimilation
process or a more sophisticated nudging method could improve the forecast spread and skill.

The downscaled forecasts were forced at the ocean surface with daily mean atmospheric data from the global SPEAR retrospective forecasts. Each of the 10 downscaled ensemble members was forced with the corresponding ensemble member from the SPEAR reforecasts. The forcing was derived directly from the SPEAR data; no correction for bias or drift was applied before downscaling. Preliminary experiments suggested that attempting to correct biases in the forcing yielded negligible
changes in the downscaled forecast skill. The global SPEAR forecast simulations use a scheme that applies adjustments to the ocean tendencies based on typical analysis increments, and this reduces drift in the global model bias during the course of the forecast (Lu et al., 2020). A standard correction for bias and drift was applied to the downscaled forecasts after running the simulations (Section 2.3).

The regional MOM6-NWA12 model also requires ocean open boundary conditions, river discharge, and tidal forcing. Tidal
forcing at the open boundaries and by the tidal potential, both of which can be accurately predicted far into the future with basic astronomy, were applied for the forecasts as in Ross et al. (2023). For river discharge, which is substantially more challenging



to forecast, we neglected any potential predictability and instead used a smoothed daily climatology of river discharge in the forecasts. The climatological mean for each day of the year was determined for each grid point in the 1993–2019 river forcing used in Ross et al. (2023), which was derived from the GloFAS reanalysis (Harrigan et al., 2020). The climatologies were then
smoothed with a $\pm$ 5 day triangular filter following Pegion et al. (2019) and Ross and Stock (2022).

The non-tidal open boundary conditions were also specified with monthly climatologies averaged over 1993–2019 from the daily GLORYS12 reanalysis temperature, salinity, velocity, and sea level data used in the analysis simulation. Climatology was used for the boundary data under the assumption that propagation of any predictable temperature or salinity signals from the model boundaries to the coastal and shelf regions of interest would occur over timescales longer than the 1-year length
of the forecasts. For example, the propagation of anomalies from where the continental shelf intersects the northern model boundary to the Scotian Shelf and Gulf of Maine has been found to occur on time scales of 1–2 years (Brickman et al., 2018; Gonçalves Neto et al., 2021; New et al., 2021). Preliminary experiments confirmed that the forcing used at the open boundaries made a negligible difference to the forecast skill, as long as the forcing did not disrupt the Gulf Stream pathway. In addition to avoiding issues related to the Gulf Stream, using climatological boundary data also greatly simplifies the setup of the forecast
simulations. Attempts to extend the forecasts beyond 1 year of prediction, however, would very likely require using the parent model forecasts as boundary conditions or devising some other way to allow predictable signals to enter through the boundary. The same is true for developing the forecasts to predict faster propagating sea level variations.

## 2.3   Forecast post-processing

From monthly mean time series of both the downscaled NWA12 and parent SPEAR model output, we calculated area-weighted
averages of surface and bottom temperature and surface salinity within the six marine ecoregions shown in Figure 1: three Large Marine Ecosystems (LMEs) along the U.S. East Coast (Scotian Shelf, Northeast U.S. Continental Shelf, and Southeast U.S. Continental Shelf) and three areas for the North and South Gulf of Mexico and the West Florida Shelf derived from corresponding Marine Ecoregions of the World (Spalding et al., 2007). For a few analyses, we also calculated averages within two smaller Ecological Production Units (EPUs) for the Gulf of Maine and Mid-Atlantic Bight. To ensure that model area
averages were consistent with conditions on the shelf, the regions used for averaging were further reduced to include only model points with depths 200 m or shallower in the three shallow Marine Ecoregions and 600 m or shallower in the remaining LMEs and EPUs.

A climatology for each initialization month (March, June, September, and December), lead month (0–11 months), variable, and region or grid cell was calculated using the ensemble mean of the full 1993–2021 set of forecasts. Forecast anomalies were
calculated by subtracting the respective climatology from the forecasts. When compared with observed anomalies, the forecast anomalies effectively have the lead-dependent bias removed. Although this approach is commonly used, calculating the model climatology with the full set of forecasts will introduce some artificial skill by incorporating information about the future that would have been unknown in the past (Risbey et al., 2021); however, we emphasize that this information is only used for the correction of bias and drift.




## 2.4 Forecast evaluation

To evaluate the forecasts, we primarily relied on data from the GLORYS12 ocean reanalysis to represent the observed conditions. This high-resolution reanalysis compares favorably with in-situ observations relative to other reanalysis products (Amaya et al., 2023; Carolina Castillo-Trujillo et al., 2023) and has the advantage over purely observed datasets of providing complete coverage over time and space. We will note, however, that the initial conditions for the downscaled forecasts were derived by nudging towards the GLORYS reanalysis, which may confer a skill advantage to the downscaled forecasts. In this paper we will often refer to the GLORYS reanalysis as "observations" because it has been extensively evaluated regionally and to indicate where methodologically another observational or state estimate dataset could be used. As with the forecasts, monthly climatologies were calculated for each variable in the GLORYS dataset during the 1993–2021 time period and were subtracted from the data to produce anomalies, and spatial averages were calculated within the different marine ecoregions in Figure 1.

The downscaled retrospective forecasts produced by MOM6-NWA12 and the corresponding parent forecasts produced by SPEAR were compared with the reanalysis dataset using the Pearson correlation coefficient and the mean bias (the mean difference between the model and the observations). To determine how the forecast skill varied with the time of year when the forecast was initialized and the lead time, we calculated these metrics separately for each initialization month and lead time.

To quantitatively test the hypothesis that downscaling produces an increase in forecast skill, we compared the correlation coefficients from MOM6-NWA12 and SPEAR using the method of Steiger (1980). This method tests the difference between two correlation coefficients obtained from two sets of forecasts that are compared against a common set of observations and are correlated with each other. Neglecting this commonality between the two forecasts would result in overly conservative estimates of significance (Siegert et al., 2017). We used this to conduct a two-sided test with a null hypothesis that the difference between the correlation coefficients for NWA12 and SPEAR was 0. For a fair comparison between the two models, all evaluations of the SPEAR forecasts were based on only the first 10 ensemble members that were downscaled with MOM6-NWA12.

To determine if the model forecasts outperform a simple forecast, we also used the Steiger (1980) test to compare the observation—model correlations with the correlations between the observations and a set of persistence reference forecasts. The persistence forecasts were computed by taking the reanalysis monthly mean anomaly from the month before each forecast was initialized and assuming that this anomaly would remain constant throughout the forecast.

For a second comparison of the skill of the parent and downscaled model forecasts, we used the DelSole and Tippett (2016) random walk test to assess the difference in the absolute error of the two models (the absolute value of the difference between the model forecast and the observation). This test uses a time series of historical forecasts from two models. Starting at 0 at the beginning of the time series (year 1993 here), it adds 1 each time the first model (NWA12) has a lower absolute error than the second (SPEAR), and subtracts 1 each time the second model has a lower absolute error than the first. This produces a time series of the cumulative sign of the errors, and a consistent trend in the time series indicates consistent outperformance by one model. Changes in relative performance can be indicated by changes in the trend of the time series. Under the null hypothesis that the model mean absolute errors are equal, the time series follows a random walk with a probability distribution given by DelSole and Tippett (2016). For this test, unlike the other comparisons, we pooled the forecasts over all initialization times. To





simplify the presentation of the results, the forecasts from both models were resampled to seasonal (3-month) averages, and
the test was run for the resulting 4 seasonal lead times.

Finally, to assess whether the ensemble members from both models provide reliable probability information (i.e., whether
the typical ensemble spread is consistent with the typical error of the ensemble mean), the average spread of the ensemble
members was compared with the root mean square error (RMSE) of the ensemble mean forecast. If the ensemble is well-
calibrated, these two metrics will be equal, while underdispersion and overconfidence will be indicated by a larger RMSE and
overdispersion and underconfidence by a larger spread (Fortin et al., 2014). The RMSE of the ensemble mean was calculated
using the anomalies, which removes the error due to mean bias. The ensemble spread was calculated as the square root of the
average ensemble variance using Equation 16 of Fortin et al. (2014):

$$S = \sqrt{\frac{1}{T}\sum_{t=1}^{T} s_t^2} \tag{1}$$

where $s_t^2$ is the unbiased estimate of the variance of the ensemble:

$$s_t^2 = \frac{1}{E-1}\sum_{e=1}^{E}(\overline{X_t} - X_{t,e})^2 \tag{2}$$

and $T$ is the number of forecast times, $E$ the number of ensemble members, and $X$ the ensemble of forecasts.

## 3   Results

Sea surface temperature prediction skill for both the global SPEAR and downscaled NWA12 models outperforms a baseline
forecast of persistence across many initialization months, lead times, and LMEs (Figure 2). The highest correlations are gen-
erally found in the Northeast U.S. LME, where both NWA12 (Figure 2d) and SPEAR (Figure 2e) exhibit a diagonal pattern
of high skill corresponding to forecasts of autumn and winter SST anomalies. This pattern is consistent with the reemergence
of early spring temperature anomalies from below the summer mixed layer in the autumn and winter (Section 4.1). NWA12
substantially improves on the SST prediction skill of the SPEAR model in the Northeast U.S. LME (Figure 2f). This improve-
ment is highest for forecasts initialized in September and at later lead times in June and December, where the SPEAR forecasts
have correlations that are sometimes negative and worse than persistence. A possible cause of this error will be discussed in
Sections 4.1.2–4.1.3. Both models also have high forecast correlations in the Scotian Shelf LME (Figure 2a–c), although with
a less clear pattern to the skill and few statistically significant improvements in the downscaled model. In the remaining four
southern LMEs (Figure 2g–r), the correlation coefficients are comparatively lower than those in the SS and NEUS and the
difference between the two models is negligible. Most forecasts have a higher correlation than persistence, however.

Forecast correlation coefficients are higher for bottom temperature (Figure 3), which partially reflects its increased persis-
tence. Most downscaled forecast correlations are higher than the persistence correlation, however, though the majority are not
significantly higher. In the Northeast U.S. (Figure 3d–f), the pattern of forecast skill and downscaling improvement is similar to
that seen for surface temperature. In other regions, NWA12 bottom temperature predictions have significantly higher skill than




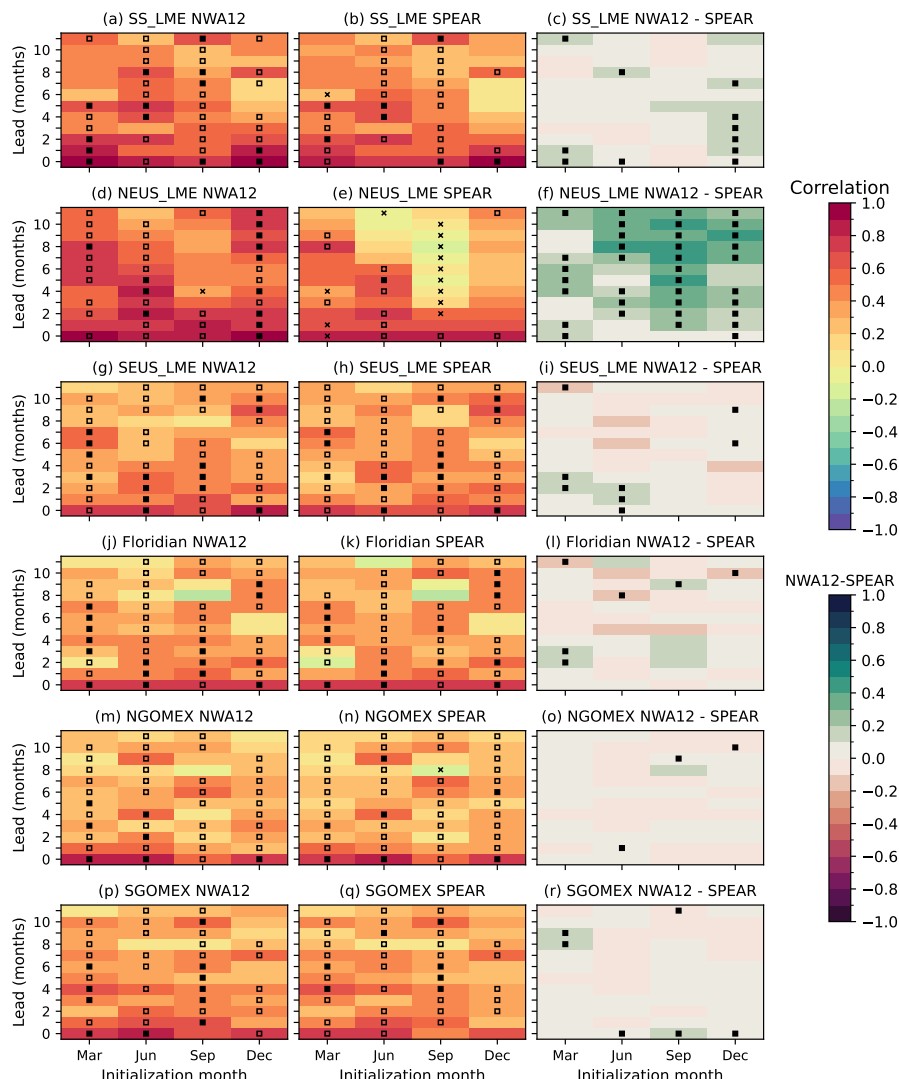

**Figure 2.** Comparison of sea surface temperature forecast–observation correlation coefficients for NWA12 (left column) and SPEAR (middle column) and the difference between the two (right column). In the left two columns, an open black square indicates a correlation that is greater than the persistence–observation correlation, a filled black square indicates that this difference is statistically significant, and a black x indicates a correlation that is significantly less than the persistence–observation correlation. In the right column, a filled black square indicates that the anomaly correlation of NWA12 is significantly different than the anomaly correlation of SPEAR.





SPEAR for some cases where they did not for surface temperature. Skill and improvement on the Scotian Shelf (Figure 3a–c)
resembles the Northeast U.S. sea surface temperature pattern of skill for forecasts verifying in autumn and winter. This pattern
only appears in the downscaled predictions; in the global model predictions, some of the anomaly correlations are significantly
less than persistence. In the Southeast U.S. (Figure 3g–i), the downscaled model has skill greater than persistence and SPEAR
across a wide range of times, except in the winter, which may reflect poor prediction of the mixing of surface water to the
bottom. A similar pattern of limited winter improvement in the downscaled model is seen in the Northern Gulf of Mexico
(Figure 3m–o). However, the three LMEs along the shores of the Gulf of Mexico (Figure 3j–r) have improved downscaled
forecast skill for forecasts verifying in the late summer and early autumn.

The patterns of correlation seen in surface and bottom temperature also generally appear in the correlation skill for sea
surface salinity (Figure 4). For example, the NEUS (Figure 4d–f), Floridian (j–l), and NGOMEX (m–o) regions have a general
pattern of higher correlation for forecasts verifying in the autumn and winter. Similarly, the SEUS LME (g–i) has lower skill
for predictions of winter, consistent with lower skill of winter bottom temperature and the possible connection to unpredicted
mixing. In most regions, the improvements of the downscaled NWA12 over the global SPEAR are higher for surface salinity
than for temperature. Because the downscaled forecast river discharge is set to climatology, and precipitation and evaporation
are poorly predicted and/or negligible drivers of salinity, the forecast skill here must primarily come from better resolution
of salinity anomalies in the initial conditions and better prediction of their advection and dispersion (Section 4.1.2), or better
prediction of salinity advection by anomalous currents.





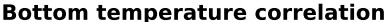

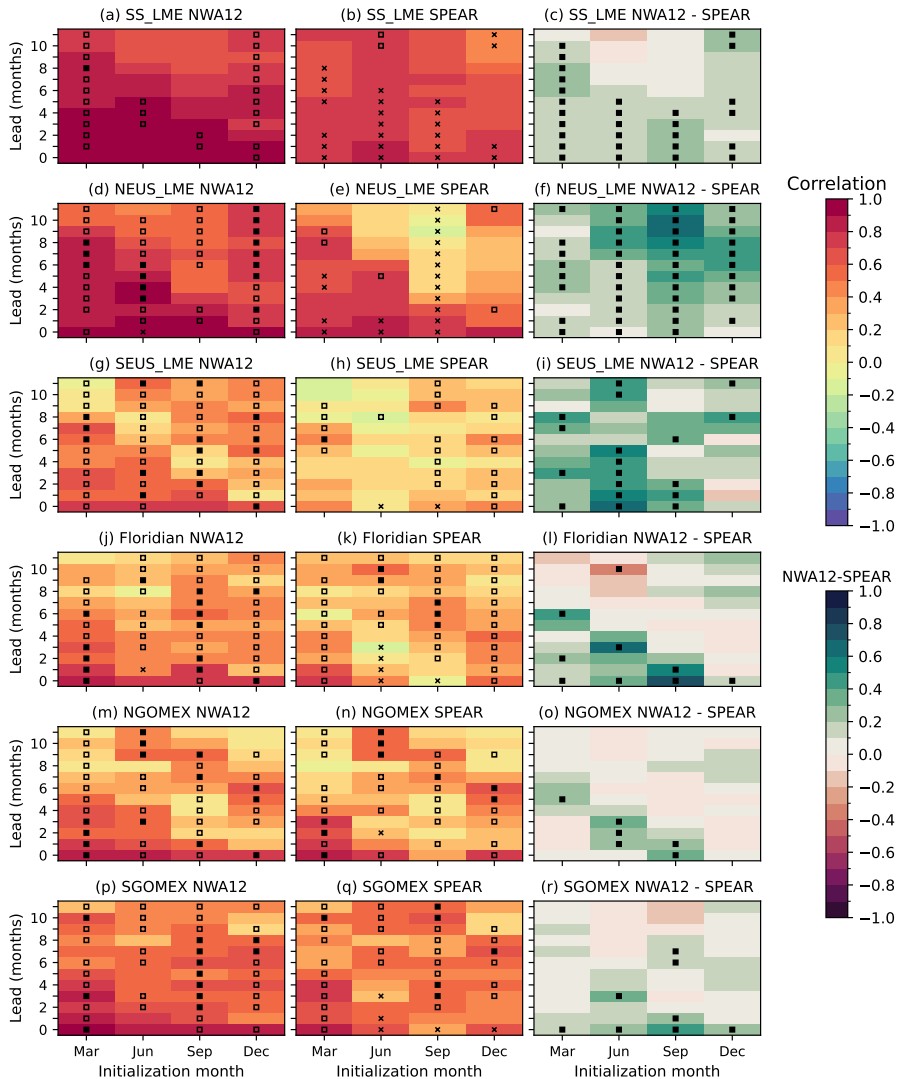

**Figure 3.** Comparison of bottom temperature forecast–observation correlation coefficients for NWA12 (left column) and SPEAR (middle column) and the difference between the two (right column). In the left two columns, an open black square indicates a correlation that is greater than the persistence–observation correlation, a filled black square indicates that this difference is statistically significant, and a black x indicates a correlation that is significantly less than the persistence–observation correlation. In the right column, a filled black square indicates that the anomaly correlation of NWA12 is significantly different than the anomaly correlation of SPEAR.



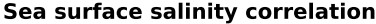

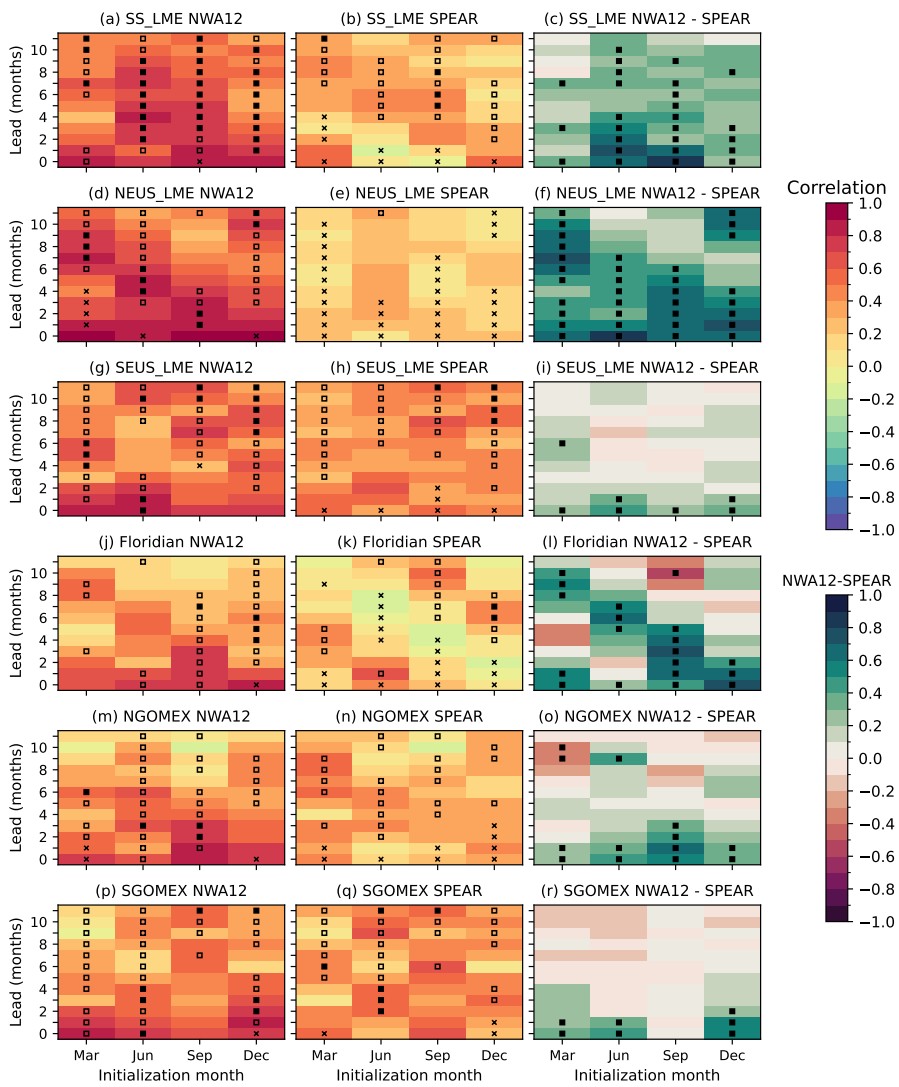

**Figure 4.** Comparison of sea surface salinity forecast–observation correlation coefficients for NWA12 (left column) and SPEAR (middle column) and the difference between the two (right column). In the left two columns, an open black square indicates a correlation that is greater than the persistence–observation correlation, a filled black square indicates that this difference is statistically significant, and a black x indicates a correlation that is significantly less than the persistence–observation correlation. In the right column, a filled black square indicates that the anomaly correlation of NWA12 is significantly different than the anomaly correlation of SPEAR.





For SST, the mean bias in the SS and NEUS LMEs is typically worse in NWA12 than in SPEAR (indicated by purple colors in the right column of Figure 5), while improved biases are often found in the southern regions (green colors in the right column of Figure 5). For the SS and NEUS LMEs, the bias has worsened in NWA12 despite the improved anomaly correlation. In these two regions (Figure 5a,d), NWA12 has a pronounced cold bias that develops in spring and dissipates in summer. This bias, as

well as the warm autumn bias, appear to be inherited from the global model, although the global biases are generally shifted 1–3 months earlier. Over all times, NWA12 is generally cooler than SPEAR, which is apparent by how NWA12 worsens the cool biases in SPEAR and improves the warm biases. It should be noted that SPEAR includes an ocean tendency adjustment (Lu et al., 2020) which helps keep biases in check during the course of the forecasts. The benefit of this adjustment is also evident from the lack of bias drift over lead times in both models—forecast biases depend mostly on verification date, not on

lead time. Finally, whereas SST correlation in the southern three LMEs was similar in NWA12 and SPEAR, NWA12 typically has lower SST bias in these regions.

Bottom temperature biases also show a strong dependence on verification month, except in the Southeast U.S. region (Figure 6). NWA12 generally has a cool bias in the two northern LMEs and a warm bias elsewhere. The cool bias develops in winter and persists for the remainder of the forecast (note that June initializations in Figure 6a,d have near-zero bias for summer and

autumn, but in December initializations the following summer and autumn are biased cool). In the southern three LMEs (Figure 6j,m,p), NWA12 has a warm bias in summer and autumn, but this bias does not persist into the winter. NWA12 has substantially less bottom temperature bias than SPEAR in the Southeast U.S. LME (Figure 6g–i), which is narrow and dominated by the Gulf Stream. In other regions, the differences between the global and downscaled prediction biases are more mixed.

Some of the largest and most consistent improvements from downscaling are found in the reduced sea surface salinity biases

(Figure 7). The reduced bias is most notable in the NEUS (Figure 7d–f) and NGOMEX (m–o) LMEs, both regions of high river discharge and sharp coastal salinity gradients, where SPEAR underestimates the mean surface salinity. Across all regions, the downscaled model generally has a mild salty bias, in contrast to SPEAR's fresh biases. It should be noted that the GLORYS12 reanalysis used as the observations in this comparison does not simulate salinity as well as it does temperature (Amaya et al., 2023; Carolina Castillo-Trujillo et al., 2023), and some of the reduction in bias may be due to the use of this reanalysis in the

derivation of the initial conditions for the downscaled forecasts.

Aggregated across all four initialization months and averaged into 3-month seasons, the random walk metric shows that NWA12 SST anomalies have a statistically significant lower mean error than SPEAR (positive random walk value) for all lead seasons in the Northeast U.S. (Figure 8b) and all but the last season in the Floridian region (d). Overall, the mean absolute error (MAE) of predicted SST anomalies in the Northeast U.S. is approximately 20% lower in the downscaled NWA12 model

than in the global model. NWA12 also has significantly lower error for the first season in the Southeast U.S. (Figure 8c) and marginally statistically significant improvements for the Southern Gulf of Mexico in seasons 0 and 2 (f), although in these cases the magnitude of the improvement in the overall MAE is small. None of the tests conclude that the error of NWA12 is significantly worse (i.e., no final value of the test lies below the 90% confidence interval for a random walk with 0 mean), although the last season is close in the Southern Gulf of Mexico and was occasionally significant in the NGOMEX. For the

Northeast U.S. and, to a lesser extent, the Scotian Shelf LMEs, the results suggest a possibility that the downscaled forecasts



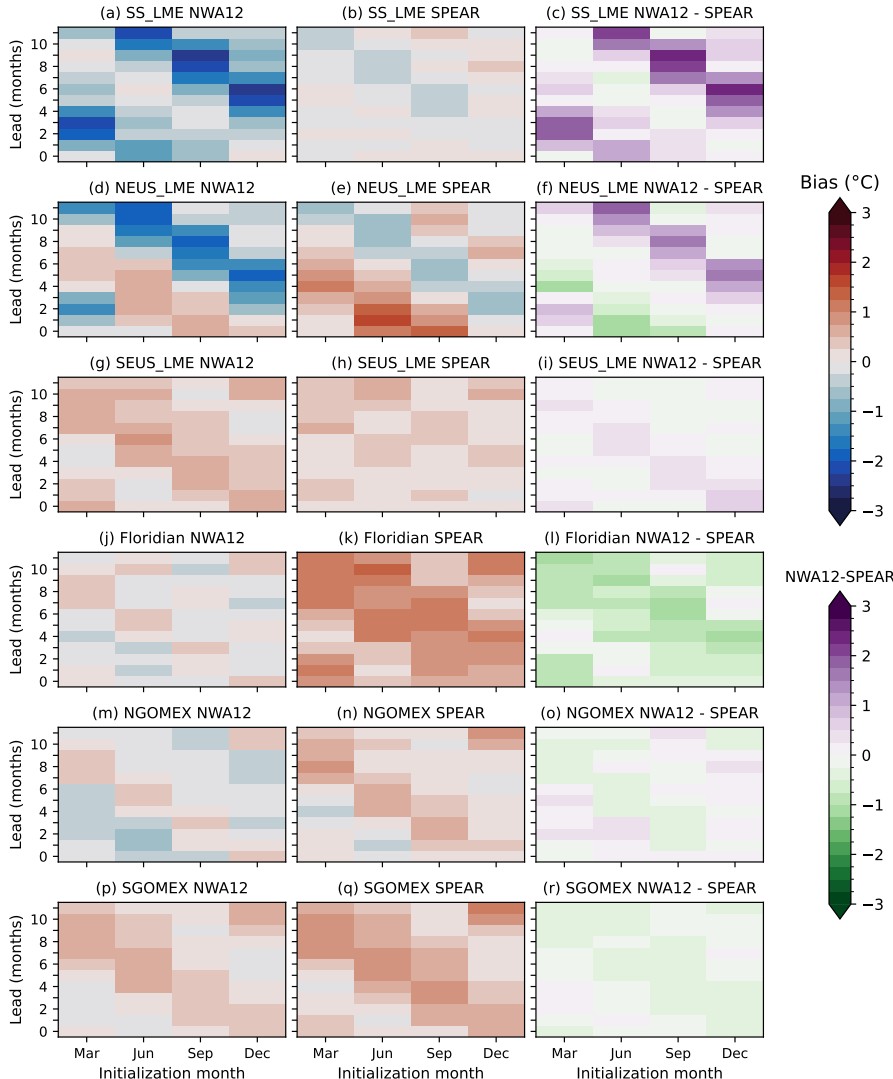

**Figure 5.** Comparison of sea surface temperature forecast–observation mean bias for NWA12 (left column) and SPEAR (middle column) and the difference between the magnitude of the mean bias in NWA12 and SPEAR (right column). Green shades in the right column indicate that NWA12 mean SST is closer to the observed mean than SPEAR, while purple shades indicate that the SPEAR mean is closer than NWA12.

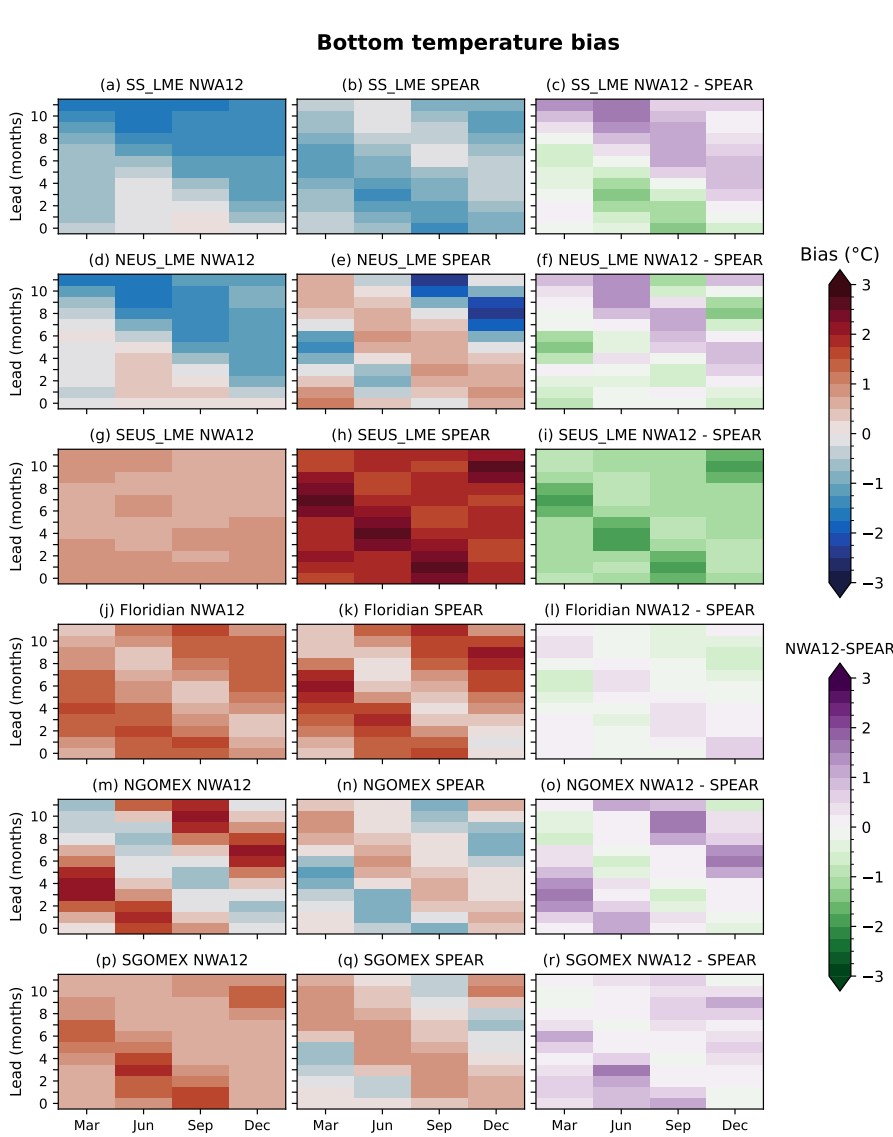

**Figure 6.** Comparison of bottom temperature forecast–observation mean bias for NWA12 (left column) and SPEAR (middle column) and the difference between the magnitude of the mean bias in NWA12 and SPEAR (right column). Green shades in the right column indicate that NWA12 mean bottom temperature is closer to the observed mean than SPEAR, while purple shades indicate that the SPEAR mean is closer than NWA12.



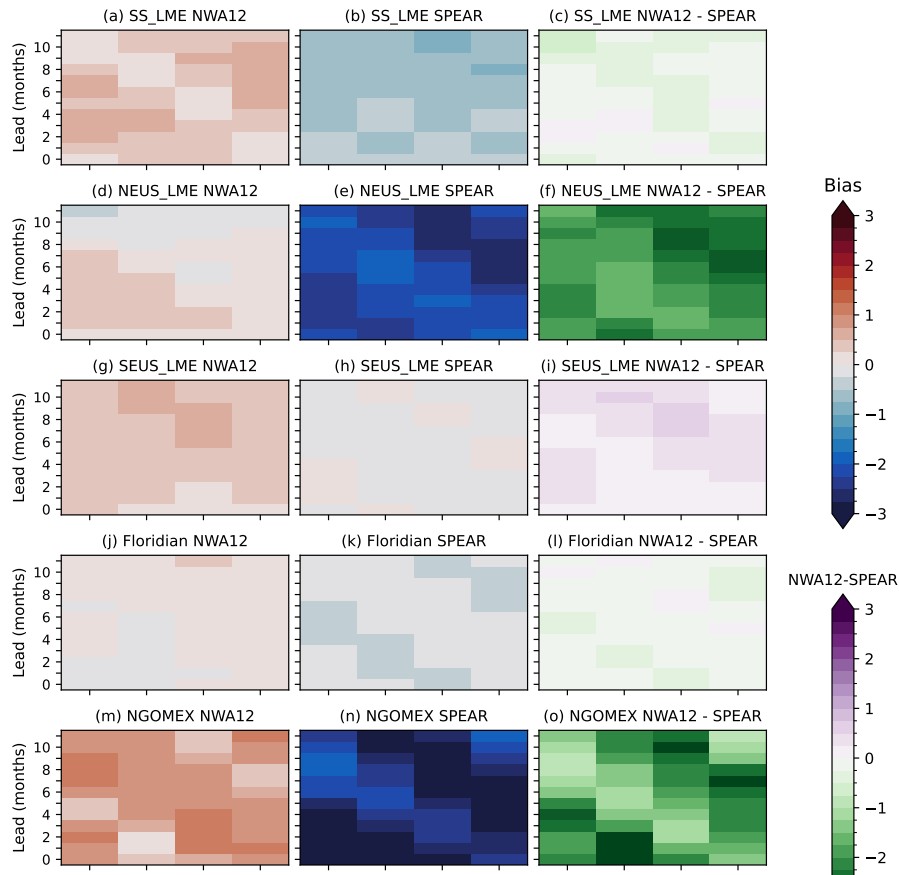

**Figure 7.** Comparison of sea surface salinity forecast–observation mean bias for NWA12 (left column) and SPEAR (middle column) and the difference between the magnitude of the mean bias in NWA12 and SPEAR (right column). Green shades in the right column indicate that NWA12 mean surface salinity is closer to the observed mean than SPEAR, while purple shades indicate that the SPEAR mean is closer than NWA12.





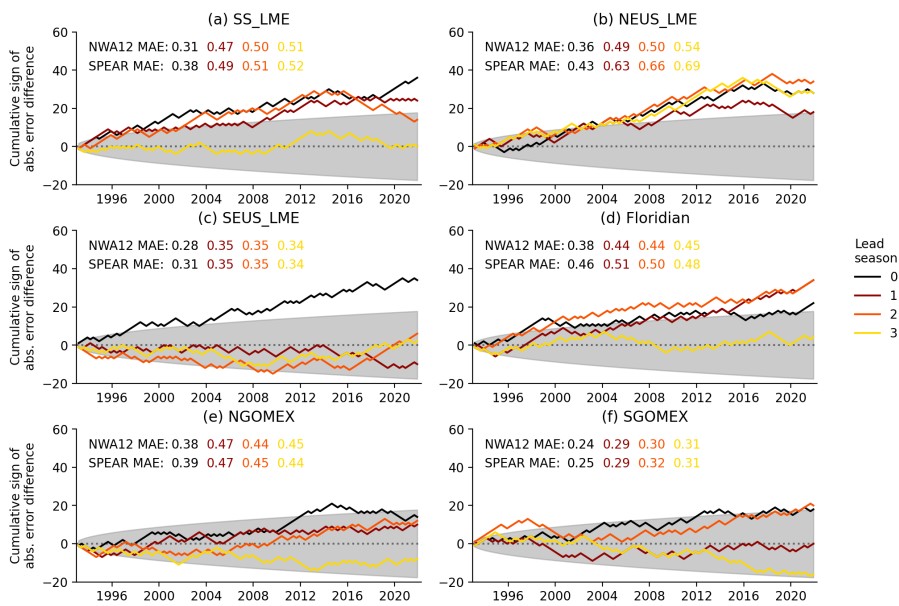

**Figure 8.** Cumulative sign of the difference of the absolute errors of SST forecasts from NWA12 and SPEAR. Forecasts have been aggregated into 3-month seasonal averages. Gray shaded region indicates where the difference in absolute error between the two models is not statistically significant at $\alpha = 0.1$. Annotations in the top left of each panel give the overall mean absolute error (MAE) of the model forecasts for each lead time.

were no longer better than the global model forecasts after around 2016 (indicated by roughly flat lines); however, the sample size is too small for this result to be conclusive.

Despite the use of initial conditions that were identical for each of the 10 ensemble members, the downscaled SST forecasts have similar or better ensemble spread characteristics than the global model (Figure 9). In all regions, both models have similar

ensemble spread in the first forecast month, which indicates that the ensemble of atmospheric conditions provided by SPEAR is able to rapidly force the downscaled ensemble members to diverge from their identical initial conditions. Across most lead times in the two northern LMEs (Figure 9a–b), NWA12 has both higher ensemble spread and lower RMSE than the SPEAR forecasts, although the RMSE generally remains greater than the ensemble spread (i.e., NWA12 remains overconfident, but less so). Differences between the two models are smaller in the remaining three regions, aside from the increased spread in the

Southeast U.S. and decreased spread and RMSE in the Floridian region in NWA12. Both models are also overconfident (RMSE greater than spread) in the Southeast U.S. and Northern Gulf of Mexico regions. In the southern Gulf of Mexico LME, only the first forecast month is substantially overconfident. However, the RMSE of both models rapidly converges to the RMSE of the climatology, which indicates no forecast skill.





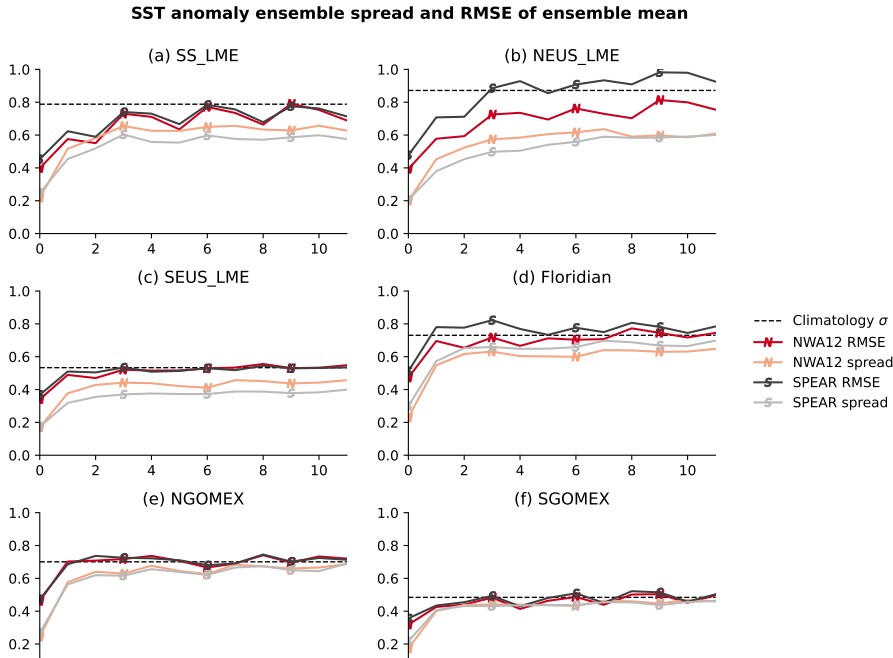

**Figure 9.** RMSE of NWA12 and SPEAR ensemble mean SST anomaly forecasts (dark red and gray, respectively), spread of the model ensemble members (light red and gray), and the standard deviation of the observed anomalies (black dashed line). Letter markers on the lines mark whether the line is for NWA12 (N) or SPEAR (S).

## 4 Discussion

In this section, we examine three mechanisms that contribute to the prediction skill and how better resolution of these mechanisms improves the skill in the downscaled forecasts: the reemergence of SST anomalies, advection of salinity anomalies, and temperature and salinity anomalies driven by variability of the Gulf Stream (Section 4.1). This examination primarily focuses on the Northeast U.S. region, where downscaling produced consistent improvements in surface and bottom temperature and surface salinity. After analyzing the mechanisms that contribute to prediction skill, we determine whether the skill would be

reduced if we accounted for the effect of the predictable trend in SST (Section 4.2). Finally, we discuss whether there is an opportunity to increase the forecast skill by increasing the number of members in the model ensemble (Section 4.3).



## 4.1 Sources of model skill and improvement from downscaling.

### 4.1.1 Reemergence of temperature anomalies

The reemergence mechanism, whereby temperature anomalies are deeply mixed during the winter, then insulated from the
atmosphere beneath the shallow summer mixed layer, and finally mixed back to the surface in the autumn and early winter as
the mixed layer deepens again (e.g., Alexander et al., 1999), is one prominent source of long-term seasonal predictability in
many regions (Jacox et al., 2020). In the results (Figure 2), we found that SST forecasts for the Northeast and Southeast U.S.
LMEs were most skillful for predictions of autumn and winter SST anomalies, even for forecasts initialized in March or the
previous December, which is consistent with predictable reemergence of SST anomalies.

To quantitatively evaluate the observed and predicted reemergence of SST anomalies, we first calculated the correlation be-
tween detrended SST anomalies in December and March and anomalies in subsequent months for the Northeast and Southeast
U.S. LMEs and two smaller Ecological Production Units (EPUs) within the NEUS LME (Figure 10). These region-average
anomalies were calculated only for areas deeper than 60 m to avoid including nearshore regions that are too shallow to develop
deep anomalies that reemerge. We also calculated the index of reemergence developed by Geiss et al. (2020), which is the
difference between the actual lagged correlation and the lagged correlation expected from a best-fit first-order autoregressive
process. Reemergence is indicated by a lagged correlation higher than expected; in Figure 10, we plot black triangles when the
difference is greater than 0.1. This index is similar to the approach developed by Byju et al. (2018), who developed an index
using the difference between lagged correlations with winter temperature in the following 6 and 12 months, except that it can
identify reemergence in any month. Here we fit the autoregressive process to data from 1–5 months after the start month by
minimizing the RMSE of the autoregression over data from these months. For the forecasts, the 10 ensemble members were
pooled together rather than using the ensemble mean.

   In the GLORYS reanalysis, reemergence is evident in the Northeast U.S. LME as a significant increase in correlation in
October or November after a minimum in summer (Figure 10a–b). Over the two EPUs within the Northeast U.S., the reemer-
gence is strongest in the Gulf of Maine (GOM; Figure 10c–d). In the Southeast U.S. LME (Figure 10g–h), March SSTs have a
similar reemergence signature, with a weaker magnitude but a longer persistence into winter.

   In the global SPEAR model, winter SST anomalies in the Northeast U.S. tend to persist for too long throughout the summer,
especially in the Mid-Atlantic Bight (MAB). March SST anomalies in the Southeast U.S. also persist for too long in the global
model, whereas December anomalies are eroded too rapidly in the summer. However, in the majority of cases the global model
forecasts still maintain a signal of reemergence (black triangles in Figure 10), although in some cases the SPEAR model
reemergence is delayed compared to the GLORYS reanalysis.

   Compared to the global SPEAR forecasts, the downscaled NWA12 forecasts generally have lagged SST correlations in
Figure 10 that are closer to the GLORYS reanalysis and are not overly persistent. Although downscaling improves the lagged
correlation, it does not completely eliminate all biases in reemergence; NWA12 is similar to SPEAR in terms of months when
significant reemergence is detected and the magnitude of the reemergence signal. For example, compared to the reanalysis,
both models have a much stronger reemergence of December SST anomalies in the mid-Atlantic Bight.





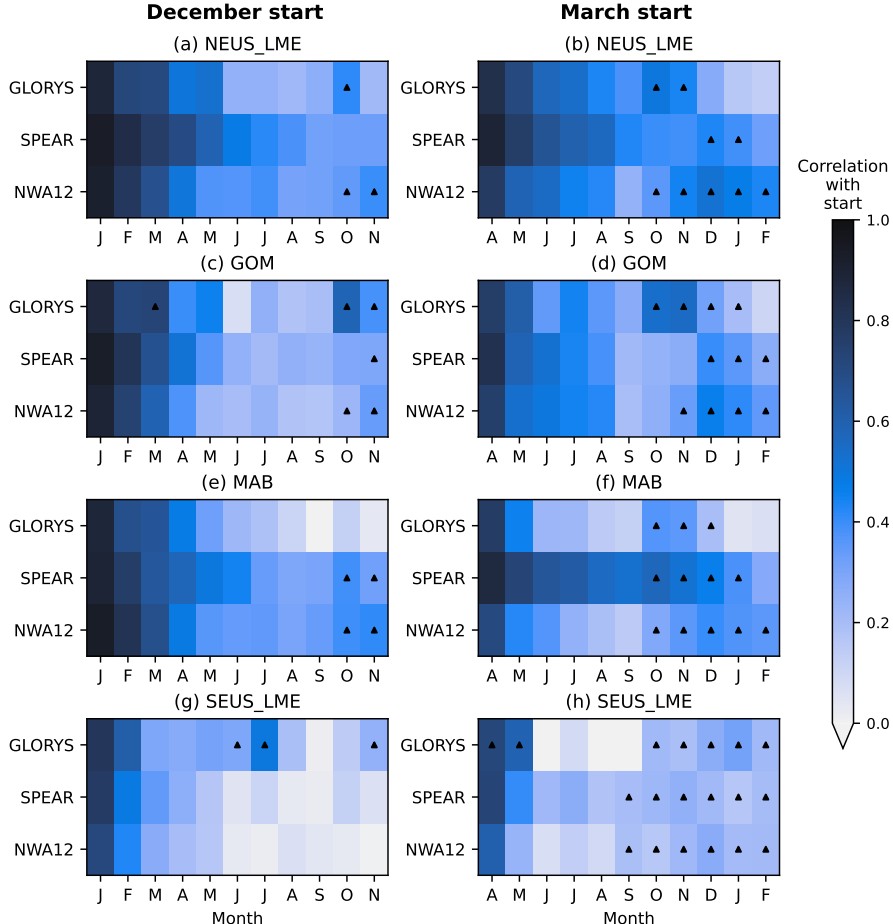

**Figure 10.** Correlations between December (left panels) and March (right panels) SST anomalies in different regions and SST anomalies in subsequent months. Black triangles are plotted where the Geiss et al. (2020) index exceeds 0.1; these triangles indicate where the correlation is greater than expected from a first-order autoregressive process.

The presence of reemergence in the forecasts is consistent with the reemergence of SST forecast skill in Figure 2, particularly for the downscaled predictions. In the Northeast U.S., the skill of forecasts initialized in both March and December peaks in the subsequent November, which matches the timing of the forecast reemergence of SST anomalies in Figure 10. In the Southeast U.S., March-initialized forecast skill has a minor reemergence in October, which matches the weak reemergence of SST in the observations and forecasts. December-initialized skill, however, has a large peak in September, which does not match any modeled or observed signal of reemergence. The most notable bias that appears in the downscaled results is a tendency for the strongest reemergence signal to appear 1–2 months late in the Northeast U.S. LME and the two EPUs.






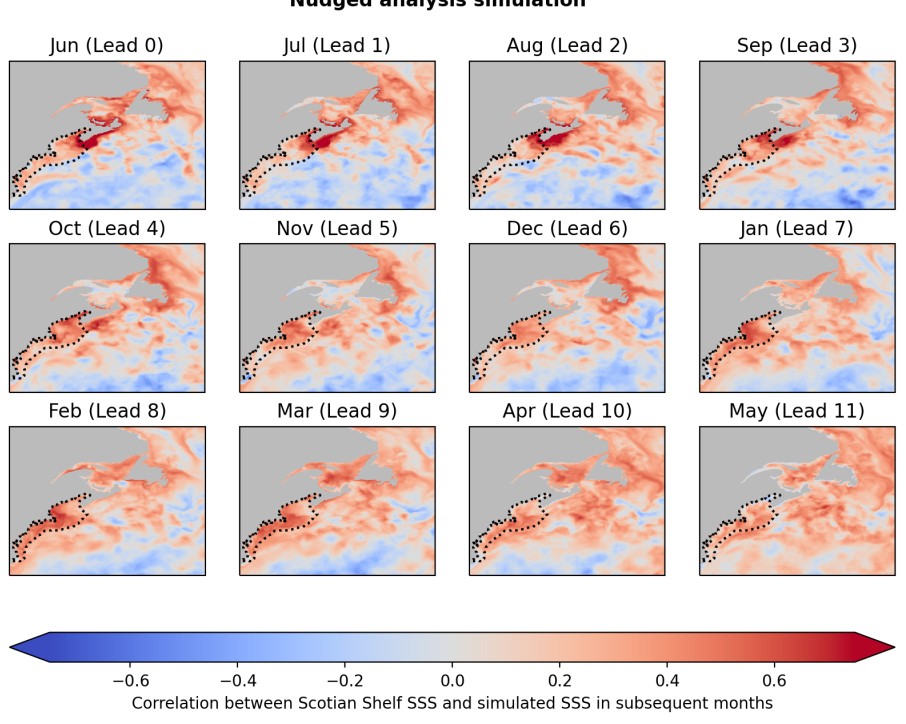

**Figure 11.** Correlation between June sea surface salinity in the Southwest Scotian Shelf and sea surface salinity in subsequent months, from the analysis simulation. The dotted line shows the boundary of the Northeast U.S. LME.

### 4.1.2 Advection of water masses

The Nova Scotia Current is a major advective pathway for cool, fresh surface water to enter the Northeast U.S. LME (Du
et al., 2021; Grodsky et al., 2017). In Figure 11, we show the correlation between June sea surface salinity averaged over a box on the Scotian Shelf just outside of the Northeast U.S. LME and sea surface salinity in following months, from the analysis simulation. This figure shows evidence that salinity anomalies starting on the Scotian Shelf are advected to the eastern Gulf of Maine in 1–2 months and continue counter-clockwise around the Gulf before reaching the Mid-Atlantic Bight after 7–9 months, which is consistent with the well-known direction and speed of the circulation in the Gulf of Maine (Smith et al.,
2001). Although the Nova Scotia Current is strongest in the winter (Han et al., 1997), we found the strongest advection signal for salinity anomalies by starting with June Scotian Shelf salinity, perhaps due to the better maintenance of surface properties during stratified summer conditions; the influence of other currents, such as transport in through the Northeast Channel, that are stronger in the spring and summer (Smith et al., 2001); or the process of advection by the cyclonic circulation in the Gulf of Maine that strengthens during the summer and autumn and reaches a maximum in December (Xue et al., 2000).
In Figure 12, we investigate whether the downscaled forecast model can simulate this advection process by performing a similar analysis, comparing lead-0 Scotian Shelf salinity from forecasts initialized in June with forecast sea surface salinity





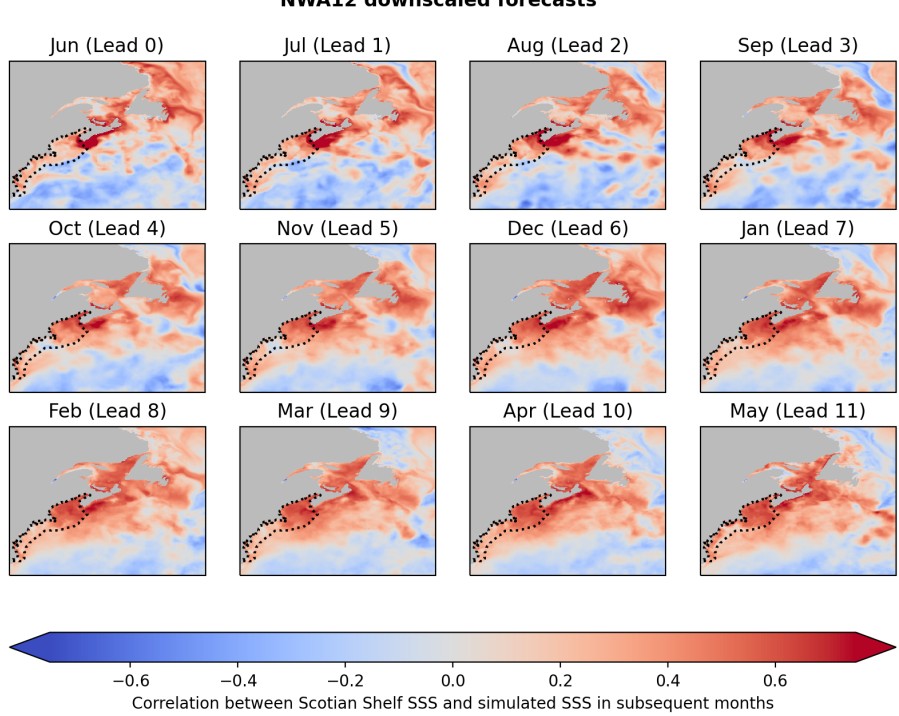

**Figure 12.** Correlation between June sea surface salinity in the Southwest Scotian Shelf and sea surface salinity in subsequent months, from downscaled forecasts initialized in June. Dotted line shows the boundary of the Northeast U.S. LME.

in subsequent months. The forecasts do capture the advection to the Eastern Gulf of Maine in the first few months. After the first few months, the advection seams to stall, with correlations remaining high in the Gulf of Maine but not increasing in the Mid-Atlantic Bight. Nevertheless, the ability to roughly capture the lagged correlations associated with this coastal advection

likely contributes to the forecast skill for surface salinity and temperature for a wide range of lead times in forecasts initialized in June (in addition to the reemergence process, which contributes to predictability around leads 3–6).




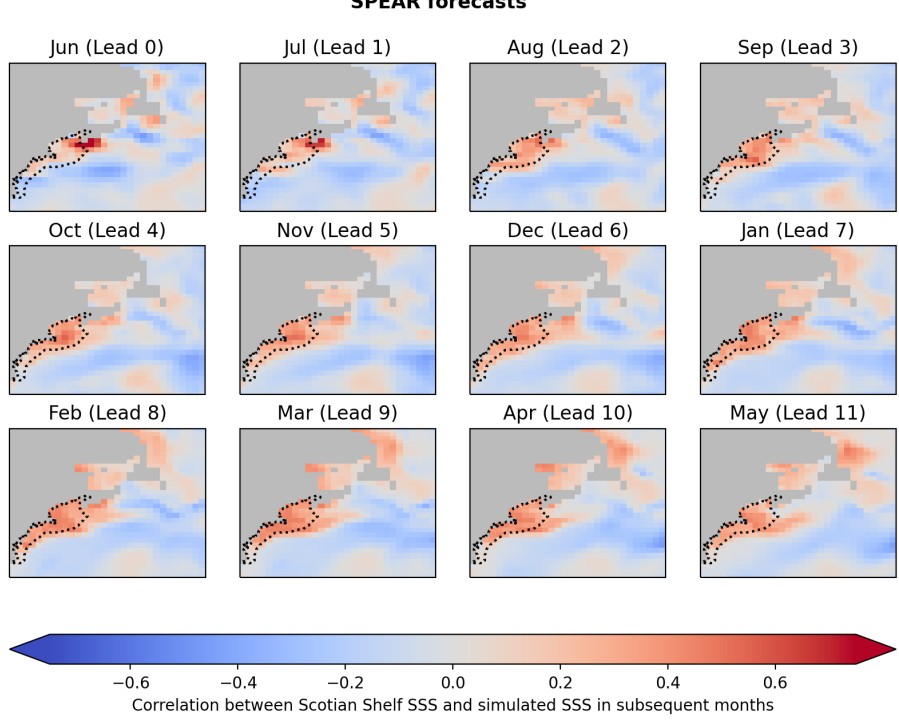

**Figure 13.** Correlation between June sea surface salinity in the Southwest Scotian Shelf and sea surface salinity in subsequent months, from SPEAR forecasts initialized in June. Dotted line shows the boundary of the Northeast U.S. LME.

By comparison, the 1° SPEAR model poorly predicts the advection of salinity anomalies from the Scotian Shelf (Figure 13). Although SPEAR does correctly predict that a salinity anomaly from the Scotian Shelf advects to the Northeast U.S. LME, the advection is too fast (note that an anomaly reaches Georges Bank by lead 3 in SPEAR versus lead 7 in the analysis simulation), and the low correlations indicate poor preservation of the advected anomaly.



### 4.1.3 Gulf Stream variability and trends driven by climate change

Over the last two decades, most of the Northwest Atlantic Ocean within the model domain has warmed substantially faster than the rest of the global ocean, and the warming has been particularly rapid along the Northeast U.S. continental shelf (Glenn et al., 2015; Pershing et al., 2015; Seidov et al., 2021; Wang et al., 2023). The rapid Northeast U.S. warming is consistent
with the effect of projected slowing of the overturning circulation associated with anthropogenic climate change (Caesar et al., 2018; Saba et al., 2016) and has been attributed to several factors, including a northward shift of the western portion of the Gulf Stream (Chi et al., 2021; Zhang et al., 2020a) and increased shedding of warm eddies (Gangopadhyay et al., 2019, 2020; Silver et al., 2023). Recent warming trends have been greatest in the summer and autumn (Friedland et al., 2023), the same seasons when the Gulf Stream is closest to the continental shelf (Du et al., 2021) and it sheds the most warm core eddies (Silver et al.,
380 2023).

In the Northeast U.S. LME, the greatest benefit of downscaling for forecasts of surface temperature and salinity was found in forecasts initialized in September and verifying over the next several months (Figures 2, 4). When broken down into smaller regions (not shown), this improvement was greatest in the Mid-Atlantic Bight region, which is nearest to the Gulf Stream. Our hypothesis is that some of the improvement from downscaling stems from improved simulation of Gulf Stream variability and
trends and the resulting impacts on surface temperature and salinity, both in the initial conditions and during the forecast.

To test this hypothesis, we begin by examining SST trends in the observations and in the two different sets of forecasts (Figure 14). For the forecasts, we evaluate the trend across initialization years from forecasts of a given lead time (or verification month); note that the trend is the same for the sea surface temperature and the lead-dependent SST anomaly. Consistent with Friedland et al. (2023), the linear warming trend in the observations during 1993—2022 was greatest during late summer and
autumn, although the confidence intervals are broad. Trends in the downscaled forecasts are slightly higher than observed and are fairly consistent regardless of initialization or lead time, again with confidence intervals that overlap the other forecasts and observations in all cases (Figure 14a). In the parent SPEAR forecasts, however, most SST trends are lower than the observations, particularly in winter and spring and in forecasts initialized in summer and autumn when the predicted SST trends are actually negative (Figure 14b). The trend is most negative in September-initialized forecasts of April SST. The
periods of erroneously low SST trends overlap with the periods of low or negative SST forecast skill in the SPEAR model (Figure 2).

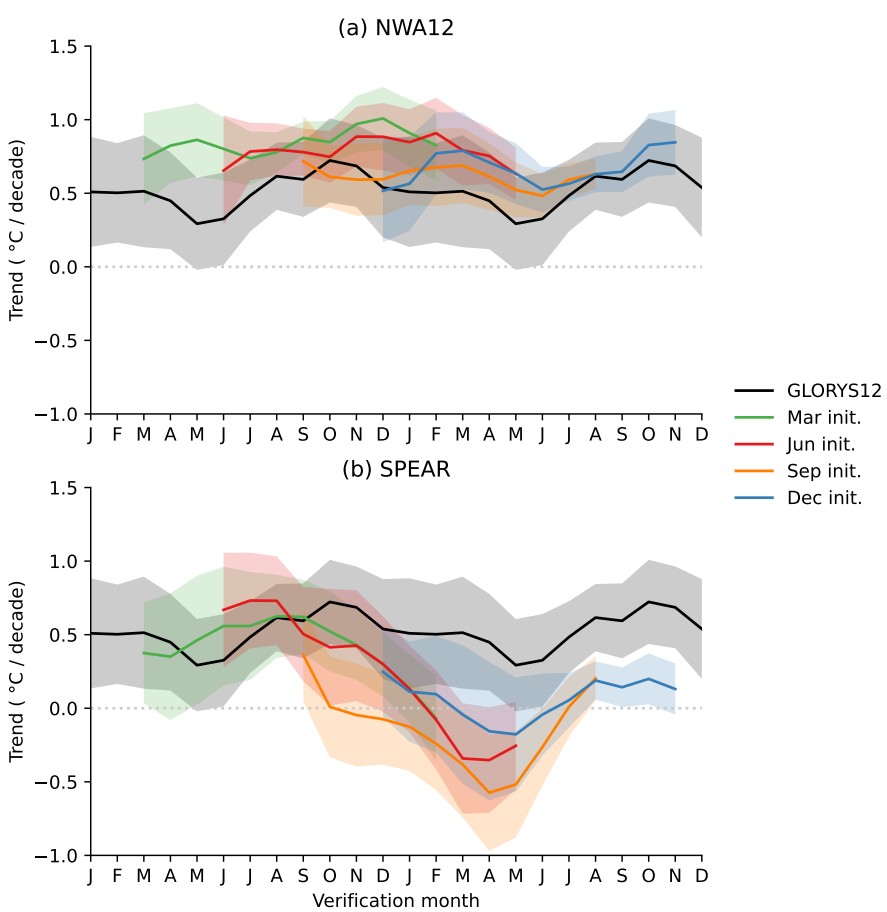

**Figure 14.** Linear trend of Mid-Atlantic Bight average SST as a function of the forecast initialization month and lead time. Shading denotes 90% confidence intervals.



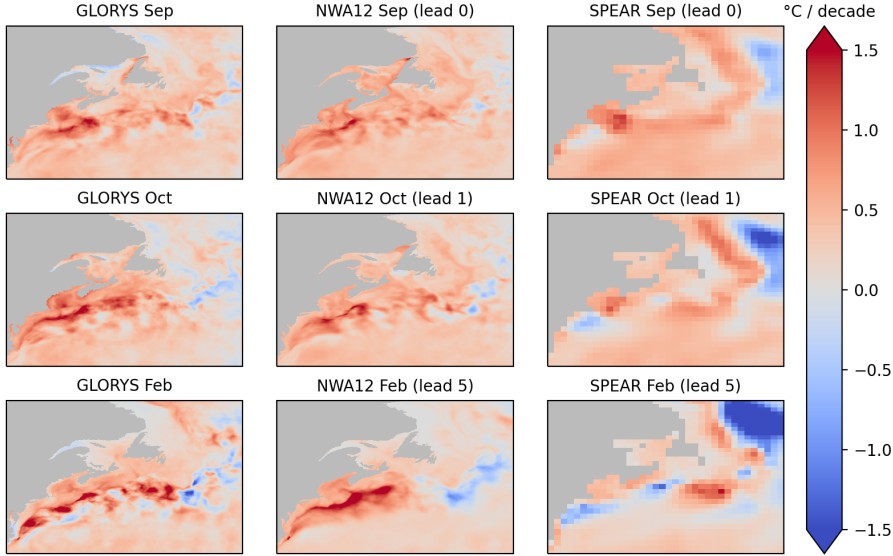

**Figure 15.** Observed linear SST trends from the GLORYS12 reanalysis in September, October, and February (left panels), SST trends in September initialized forecasts that verify in the same months from the NWA12 model (center panels) and the SPEAR model (right panels).

In Figure 15, we examine the spatial patterns of SST trends in the Northeast U.S. and Scotian Shelf regions from observations and September-initialized forecasts from the NWA12 and SPEAR models. The downscaled NWA12 forecasts have the right SST trends during the initial month and continue to correctly simulate the trends over the course of the forecasts. In forecasts of September (lead 0), the SPEAR forecasts have been correctly initialized with warming SSTs in the Gulf of Maine, but SSTs in the initial month have been incorrectly cooling in the Mid-Atlantic Bight. This cooling trend becomes worse in subsequent forecast months and extends along the shelfbreak north of the typical Gulf Stream position in February.

Finally, in Figure 16, we examine the trends of geostrophic current speeds in the AVISO satellite altimetry product provided by the Copernicus Marine Service (DOI:10.48670/moi-00148) and the two forecast models. A northward shift of the Gulf Stream is evident in the satellite observations in Figure 16a,d,g as a line of increased current speed immediately north of a line of decreased speed, particularly west of 67°W, which is both consistent with other observations (Chi et al., 2021; Zhang et al., 2020a) and simulated remarkably well in the downscaled forecasts (Figure 16b,e,h). The parent forecasts, however, have weak current speed trends that are more consistent with a modest southward shift of the Gulf Stream (Figure 16c,f,i).

These results support our hypothesis that the improved forecasts in the downscaled model partially stem from improved simulation of the Gulf Stream and its impact on variability and trends in the adjacent coast and shelf region. This ability to match the observed Gulf Stream position and variability was also found by Ross et al. (2023) in a reanalysis-forced historical simulation and reflects well on the capabilities of the NWA12 model. We emphasize that this result shows the benefits of high resolution simulations rather than a deficiency specifically with the SPEAR model. The Gulf Stream is well-known to




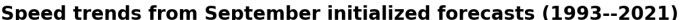

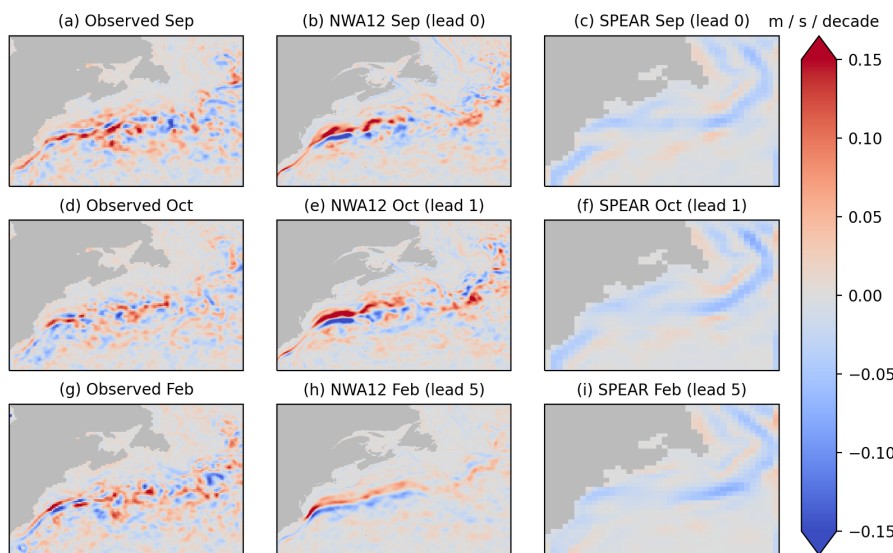

**Figure 16.** Observed linear geostrophic current speed trends from the AVISO satellite altimetry product in September, October, and February (left panels), speed trends in September initialized forecasts that verify in the same months from the NWA12 model (center panels) and the SPEAR model (right panels).

require ocean resolution of 1/10° or finer to simulate properly (Chassignet and Garaffo, 2001; Chassignet and Xu, 2017) and

the observed shift in the Gulf Stream occurred over roughly a degree of latitude (Wang et al., 2022), so there is no reason to expect that any model with an order of 1° resolution for the ocean would be able to accurately simulate the Gulf Stream shift or the resulting temperature and salinity changes.

To illustrate the benefits of resolving Gulf Stream variability at the event scale, in Figures 17 and 18 we show output from the nudged historical simulation and downscaled forecasts, respectively, for a pronounced warming event along the Northeast

U.S. and Scotian Shelf in September to December of 2020. Conditions in September featured a weak warm anomaly south of Newfoundland and west of the Grand Banks which strengthened over time and was joined by a second warm anomaly to the west in November (top row of Figure 17). Both anomalies were associated with northward excursions of the Gulf Stream (middle row) and resulted in increased temperature and salinity in the Northeast Channel (point 1, bottom left panel) and near the Scotian Shelf (point 2, bottom right). In the Northeast Channel, temperature and salinity was near average at all depths

during September (indicated by same-colored plus symbols that are near each other). As time progressed, the salinity increased at all depths, deep waters became warmer, and near surface waters cooled but much less so than usual. By December (square symbols), surface conditions were approximately 2° warmer and 0.5 units saltier than average, and water at 105 m depth was over 3° warmer and slightly less than 0.5 units saltier than average. Near the Scotian Shelf (point 2), a similar pattern was observed, with the most anomalous conditions found in October. At depth, the temperature and salinity increase from

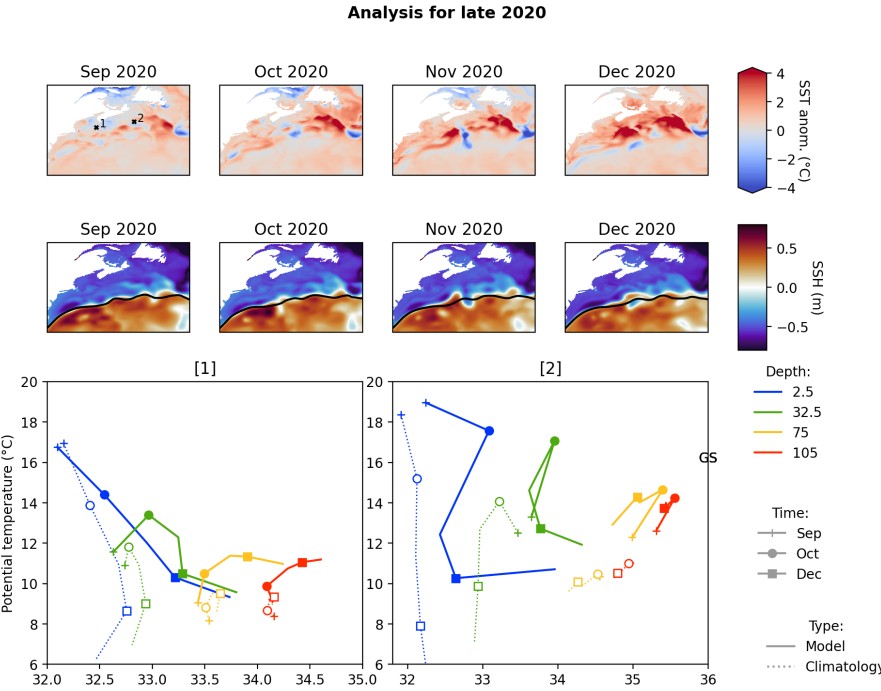

**Figure 17.** Temperature, salinity, and sea surface height during September 2020 to January 2021 from the nudged downscaled historical simulation. Top row: SST anomalies. Middle row: sea surface height, with the climatological position of the 0 cm height contoured in black. Bottom row: Temperature-salinity diagrams for the two locations marked with dots in the top left panel.

September to October was consistent with intrusion water from the Gulf Stream (marked by "GS"), and October conditions were over 4° warmer and 1 unit saltier than average.

This event was predicted remarkably well by the ensemble mean of the downscaled forecasts initialized at the beginning of September 2020 (Figure 18). The mean forecast correctly predicted the eastern warm SST anomaly in October 2020, joined by a western warm anomaly connected to a northward Gulf Stream fluctuation in November 2020. The forecast also correctly predicted increasing salinity at all levels and warming at depth in the Northeast Channel (bottom left panel) and advection of Gulf Stream-like water off the Scotian Shelf in October (bottom right). Note that this plot shows the raw forecast without the lead-dependent climatology subtracted as in other plots.

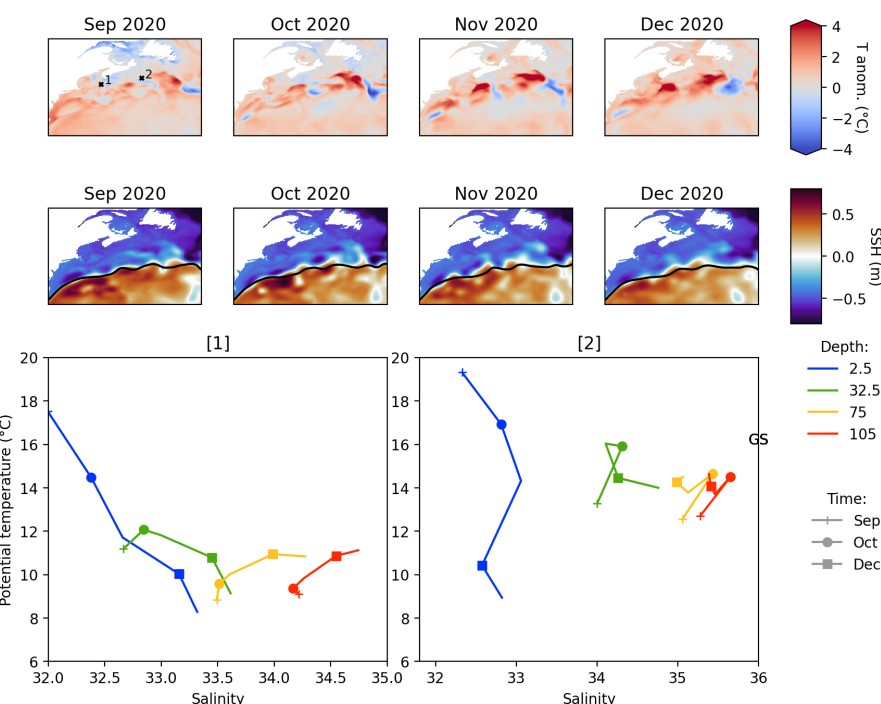

**Figure 18.** Temperature, salinity, and sea surface height during September 2020 to January 2021 from the ensemble mean of downscaled forecasts initialized at the beginning of September 2020. Top row: SST anomalies. Middle row: sea surface height, with the climatological position of the 0 cm height contoured in black. Bottom row: Temperature-salinity diagrams for the two locations marked with dots in the top left panel. The bottom row shows raw forecast values without adjustment for the lead-dependent bias.



## 4.2 Effect of long-term warming trends

Because of the rapid increase of SSTs in the Northeast U.S. region, a portion of the model surface and bottom temperature fore-
cast skill could simply come from the consistently warming initial conditions and the greenhouse gas forcing in the atmospheric
forecasts. The persistence forecast used as a baseline for comparison in the results would include the effects of warming initial
conditions, but it would not include warming that varies by season or the (likely negligible) component of the warming trend
that occurs during the course of each one year forecast. To analyze the effect that discounting the predictable trend would have
on the skill of the downscaled SST forecasts for the Northeast U.S. LME, in Figure 19 we compare the anomaly correlation for
the raw downscaled ensemble mean with the correlation after removing the linear trend from the forecasts and observations.
Trends were calculated separately for the forecasts and observations, and separately for each month and each initialization
month. The correlation of the NWA12 model predictions with the observations is reduced substantially after detrending the
forecasts and observations, especially after the first 1–2 months of lead time, indicating that the linear trend is a substantial
part of the forecast skill. However, the linear trend also contributes to the apparent skill of the persistence forecast, and the
difference between the model and persistence skill generally remains the same whether or not detrending is applied. Patterns of
correlation indicative of mechanisms behind prediction skill, such as the reemergence of skill in forecasts verifying in autumn
(Section 4.1.1) also generally remain after detrending. We also note that the trends that were removed were fit to the full time
series, which is considered an "unfair" method since this trend would not have been known in the past. Bushuk et al. (2019)
developed a "fair" method of detrending that iterates over each retrospective forecast and removes the trend calculated only
from data available before the start of each forecast. If this method was applied here, the skill of the detrended forecasts would
increase. However, we only show the unfair method of detrending since the correlation metric is also an unfair calculation
that uses means and standard deviations calculated from the full time series. Finally, we also emphasize that the origin of the
forecast skill, whether from a linear trend or dynamic variability, is likely unimportant for most marine resource management
purposes.

## 4.3 Skill versus ensemble size

In this study, we evaluated the skill of two 10-member forecast ensembles. Other high resolution ocean prediction and projec-
tion studies have typically relied on fewer ensemble members or only one prediction, often due to the computational costs of
running many ensemble members (Drenkard et al., 2021). Jacox et al. (2020) found quickly diminishing returns as the num-
ber of ensemble member forecasts of California Current System SST was increased. On the other hand, features with a low
signal-to-noise ratio, such as the NAO, may only be skillfully predicted with much larger ensembles (Dunstone et al., 2016;
Scaife and Smith, 2018; Strommen and Palmer, 2019). To assess the impact of ensemble size on seasonal forecast skill in the
NWA12 model, we created 1,000 bootstrap resamples (sampling from the 10 ensemble members for each initialization and
lead time, with replacement) of the Northeast U.S. LME SST forecasts for ensembles consisting of 1, 2, 4, and 10 members.
For each ensemble size, we calculated the median correlation coefficient for the ensemble mean and the median raw ensemble
continuous ranked probability score (CRPS; Bröcker, 2012), which evaluates the absolute error of the empirical probability

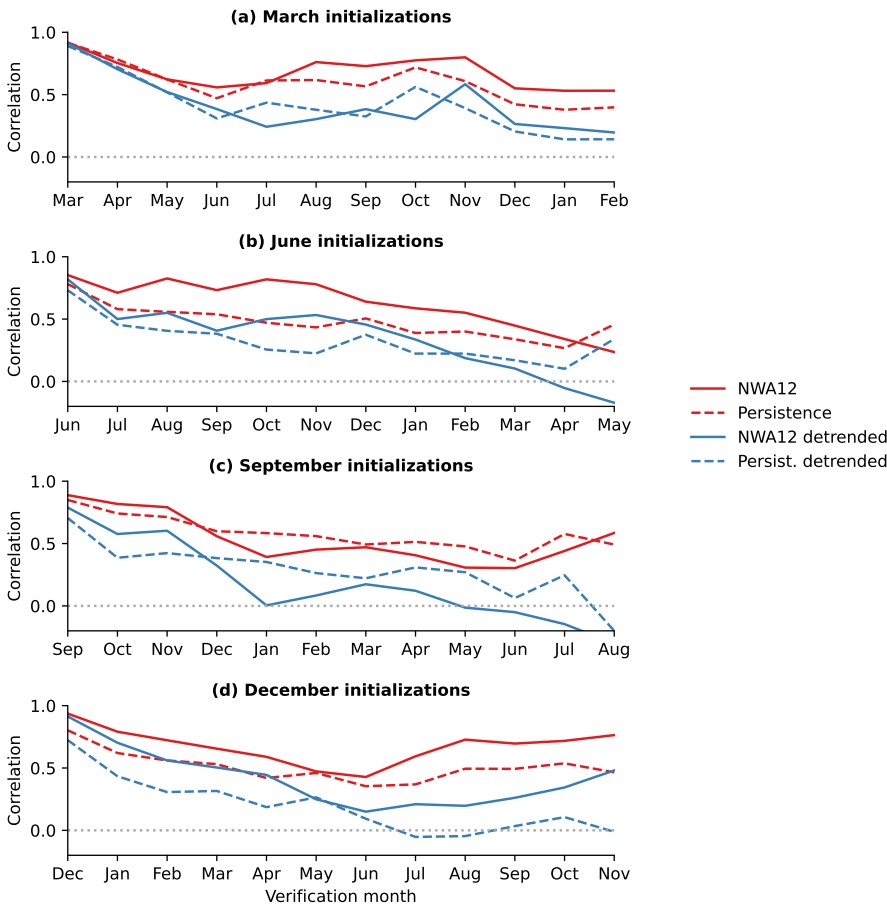

**Figure 19.** Correlation of downscaled Northeast U.S. LME SST forecasts with (red) and without (blue) detrending of the forecasts and observations, for the downscaled forecasts (solid lines) and the anomaly persistence reference forecast (dashed).





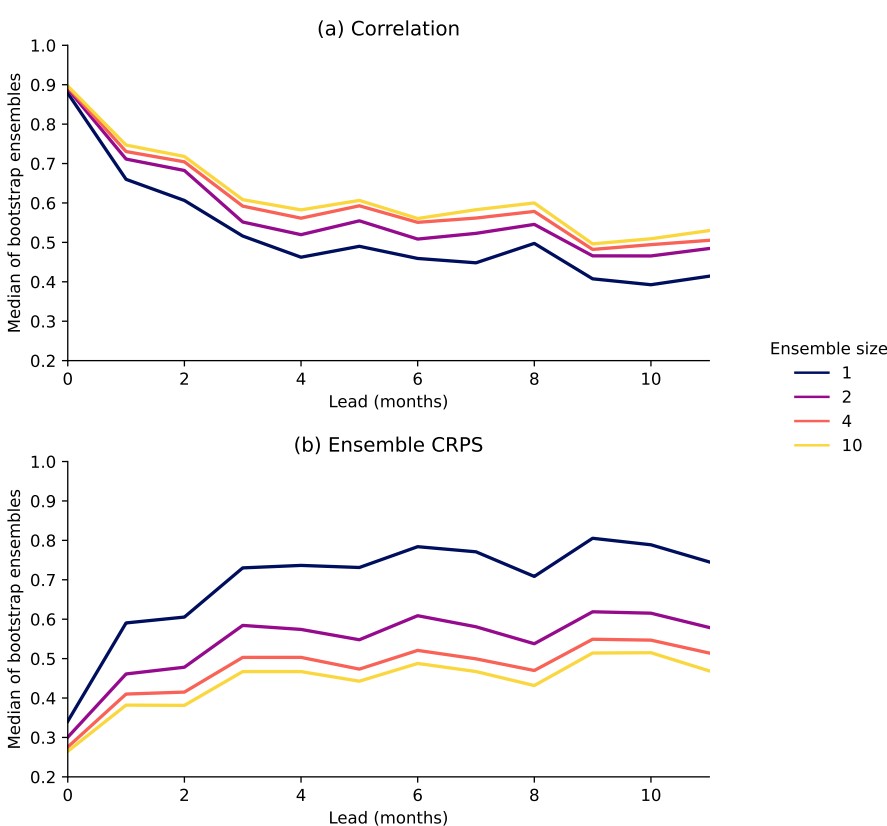

**Figure 20.** Correlation (a) and ensemble Continuous Ranked Probability Score (CRPS; b) for NWA12 as a function of the size of the model ensemble (colors). Plotted values are the medians of 1000 bootstrap samples of a given ensemble size from the full 10-member ensemble.

distribution of the ensemble. Both metrics, shown in Figure 20, indicate diminishing marginal improvements in skill (higher correlation and lower CRPS) as the ensemble size is increased; the improvement from a single member to two members is substantial, while the improvement from 4 to 10 members is minor. This suggests that, at least for predictions of monthly mean SST anomalies, an ensemble with approximately 4 members would provide a reasonable compromise between computational

costs and prediction skill. However, we expect that forecasts of rare events, such as marine heatwaves, or nonlinear variables, such as ocean chlorophyll, would still benefit from larger ensembles (Jacox et al., 2020). A larger ensemble size may also increase skill more if it was a multimodel ensemble, consisting of downscaled predictions forced by different models and/or using different models for downscaling, rather than just an ensemble of multiple runs from the same model. For example, Brodie et al. (2023) found significantly higher skill in an ensemble of 73 members from 6 different coarse resolution models

compared to a much smaller ensemble of 3 coarse or downscaled models.





# 5 Conclusions

The high-resolution downscaled seasonal predictions yielded a significant improvement to forecast skill for temperature and salinity anomalies in the Northeast U.S. Large Marine Ecosystem and maintained or improved skill in other regions along the North American East Coast. Although this initial experiment focused on predicting physical ocean conditions, the dynamical
drivers of the forecast skill suggest that skillful predictions of biogeochemical features and connections to living marine resources will also be possible. For example, similar to the reemergence of temperature anomalies that was skillfully predicted in the downscaled forecasts, Park et al. (2019) found that nitrate anomalies in the Northwest Atlantic also persist below the summer mixed layer and reemerge in the winter and lead to predictable chlorophyll anomalies in the following spring. Furthermore, the predictable advection of salinity by the cyclonic coastal circulation in the Gulf of Maine (Figures 11–12) suggests
the potential to predict the advection of harmful algal blooms by the same current (Li et al., 2014; Zhang et al., 2020b). The forecasts presented here are the first step towards building connections between predictable ocean temperature and salinity, biogeochemistry, and living marine resources along the North American East Coast to develop critically needed large scale marine ecosystem predictions (Link et al., 2023).

Although the downscaled predictions of temperature and salinity anomalies were skillful in many cases, some nontrivial
biases in the mean temperature and salinity were present even in the downscaled predictions. These ocean biases could be reduced by correcting biases in the atmospheric forcing used to drive the regional model, and experimenting with methods for correcting these biases is a goal of future research. Correcting for biases in the forcing should reduce biases in the ocean predictions but may not substantially improve the prediction skill of anomalies (as found by Jacox et al. (2023) along the U.S. West Coast). However, reducing ocean temperature and salinity biases will likely be important for obtaining accurate
predictions with coupled biogeochemical model components.

Other improvements to the basic forecast model configuration developed in this study may also be necessary to expand the skill and capabilities of the model. For example, using forecasts of river discharge, rather than a climatology of river discharge, may be necessary for predicting nearshore salinity biogeochemistry. Similarly, applying predicted conditions for the ocean open boundaries, rather than using a climatology, could be necessary for predicting quickly propagating sea level anomalies.
Finally, full data assimilation, rather than nudging towards a reanalysis, could improve prediction skill through better initial conditions and also yield better probabilistic forecasts and predictions of extreme events.

*Code availability.* The source code for each component of the MOM6-NWA12 model has been archived by Ross et al. (2023) at https://doi.org/10.5281/zenodo.7893349.

*Data availability.* All model output that was analyzed in this paper has been published at https://doi.org/10.5281/zenodo.10642294.



*Author contributions.* TD, FL, AW developed the SPEAR global prediction model. MAA contributed to the discussion on the reemergence of SST anomalies. VK contributed to the methods for forecast evaluation. ACR prepared the initial draft of the manuscript with guidance from CS. All coauthors participated in discussions during various stages of the model development and evaluation and read and approved the final version of the manuscript.

*Competing interests.* The authors declare that they have no conflict of interest.

*Acknowledgements.* We appreciate internal reviews of a draft of this manuscript by Mitchell Bushuk and Michael Jacox. This study was partially funded by a grant from the NOAA Climate Program Office. This study has been conducted using E.U. Copernicus Marine Service Information: https://doi.org/10.48670/moi-00021, https://doi.org/10.48670/moi-00148.



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
