# Peer review of "Dynamically downscaled seasonal ocean forecasts for North American East Coast ecosystems"

_EGUsphere, 2024_

## Referee Comment (RC3)

**Review of Dynamically downscaled seasonal ocean forecasts for North American East Coast ecosystems.**

**August 20, 2024**

This article discusses the skill of historical seasonal forecasts utilizing a regional North Atlantic ocean model initialized with the GLORYS12 ocean analysis (approximately the same resolution as the ocean model) and forced by atmospheric forecast conditions from the SPEAR seasonal forecast. Results are then compared with the global, low resolution, coupled forecasts of SPEAR.

I have always been an advocate for the usuage of ocean reanalysis as a tool for both initialization of forecasts – and for their use as a diagnostic tool to assess the ocean in regions where observations are sparse, or non-existant. However, the usage here, to use the GLORYS12 product as initialization for the regional seasonal forecasts – and then to assess skill against the GLORYS12 reanalysis seems somewhat incestuous to me – especially since the SPEAR ocean analysis likely differs substantially from the GLORYS12 analysis The study tnen really becomes one of assessing the initialization of a high resolution ocean model with GLORYS12 versus initializing with the ocean model component of SPEAR, and not particularly a "downscaling" of a seasonal forecast.

**My question to the authors:** If this system was to become an operational forecast system for the U.S. East Coast, would the goal be to intialize such forecasts with the real time GLORYS12 analysis *(Skill assessed in this manuscript)*, or initialized by downscaling the SPEAR ocean analysis *(Skill **not** assessed in this manuscript)*. If it is the former, then perhaps utilizing the full multi-model ensemble of NMME, as opposed to only SPEAR atmospheric component forcing, would be a more prudent approach, as that likely would increase the underdispersiveness of only using SPEAR atmospheric forecast forcing.

**Reommendation:** Despite my trepitation with regards to the skill assessment primarily against the system initialization product, **I would recommend publication after the authors answer my question and comments.**

**Itemized Comments:**

1. I believe other studies (not seasonal forecasts, however) have been undertaken with $\frac{1}{12}$th degree North Atlantic systems, although admittedly I could not find a particularly relevent study in my quick search. Perhaps the authors could more explicit with regards to the definitiion of their $\frac{1}{12}$th grid: Is the grid identical to a North Atlantic subset of the GLORYS12 grid, or how does it differ from the ORCA12 grid utilized by GLORYS?

2. It is not the responsibility of the authors to discuss the ocean initialization of SPEAR, but nonetheless, how it is initialized, and in particular, how its ocean state estimation approach differs from GLORYS12 is an important component of this study. More information is required to assess this, preferably with some explicit text in the manuscripts, but minimally by explicit citations of the SPEAR ocean initialization procedure. The manuscripts does show "0 lead" (actually 0.5 lead I believe) results that can be used to assess these differences somewhat, but some more explicit comparisons, particularly for the reemergence discussion would be useful – for instance, the manuscript shows the reemergence in the GLORYS12 reanalysis – is it also present in the SPEAR ocean analysis (or concatenation of 0 leads).

3. The statement in the conclusion, " Finally, full data assimilation, rather than nudging towards a reanalysis, could improve prediction skill through better initial conditions $\cdots$," might be true – but not necessarily when basing that skill on the reanalysis being nudged towards. This may be particularly true if not many observations are going into the ocean analysis in the areas of skill assessment, which unfortunately may be true for the coastal region, strong ocean current (short ARGO float retention) regions under study in this manuscript.

4. The spread error discussion was interesting, and the skill versus ensemble size (including CRPS results) did expand on this. But I am always interested in expanding on the probabalistic nature of the ensemble – and it would seem the reemergence diagnostics utilized here might be a natural way to expound on this. I assume the reemergence diagnostics are perform on the ensemble mean? Could an member by member diagnostics be performed that might lead to a "probabality" of re-emergence that could be accessed for skill (Brier Sc.ore)?

5. I remind the authors that Atlantic Overturning Circulation variability can be driven by atmospheric variability as well (Jackson et al, 2016; https://doi.org/10.1038/ngeo2715), with particular implications to density anomalies along western Atlantic.

6. The authors should highlight their skill assessment of ocean currents is performed using an independent "observation" source, and therefore is not as sensitive to the

initial conditions as their temperature and salinity skill assessment. Although it then may be instructive to give some evidence of current skill in the GLORYS12 analysis (Aijaz et al, 2023; https://doi.org/10.1016/j.ocemod.2023.102241)

7. In light of the previous two points, I wonder why authors did not present a more detailed skill assessment of currents beyond just a trend analysis?

8. I found the divergenging color schemes used in plots to not be particularly easy to distinguish null results, particulary with the red/green scheme used in figures 1-4 (plus I believe it is not particularly colour blind friendly). The purple/green scheme of figures 5-7 seems somewhat better – but either an explicitly white color marker for 0 difference, or 3 colour scheme (i.e. yellow as zero difference) might be preferable.

---

## Author Response (AR1)

**Reviewer #1**

**The paper provides an extensive study on downscaled retrospective forecast in the Northwest Atlantic Ocean from GFDL global model using a 1/12 configuration based on MOM6, previously designed and assessed by the Authors in another dedicated paper.**

**The methodology used for assessing the forecast is very interesting and quite comprehensive as well as the process-oriented analysis, supported by discussed results.**

We appreciate the reviewer for reading the manuscript and providing encouraging comments.

**Reviewer #2**

**I very much appreciate the opportunity to review this manuscript. This paper focuses on evaluating seasonal predictability of surface temperature and salinity and bottom temperature over the North America East Coast by using a dynamically downscaled model forecast system (MOM6-NWA12) and compared with the parent SPEAR model forecasts. Detailed discussions about sources of improved prediction skill from downscaling are included for the Northeast U.S. region, as well as discussions on the effect of long-term warming trends over this area. Besides, this paper also contributes a useful discussion on the ensemble size for reasonable prediction skill when predicting SST. This is a very important work with high quality contributing to the research field, and I only have a few minor comments on this work:**

We thank the reviewer for taking the time to read the manuscript and provide a helpful review.

**1. Some description about seasons in the Result section are confusing, not sure if authors are talking about initialization seasons or forecast seasons. For example, on lines 237-239, "the downscaled model has skill greater than persistence and SPEAR across a wide range of times, except in the winter…". It is not clear if "winter" here refers to the initialization month of December or forecast months in winter.**

We see how this could be confusing. For lines 237-239, we have reworded it to "except for forecasts verifying in December". We have also clarified in this and a few of the following paragraphs whether the months or seasons we mention are the initial or verification months.

**2. Lines 267-268, the Southeast U.S. LME, as shown in Figure 1, is not narrow compared to most other LMEs. I also question on its dominance by the Gulf Stream, as Gulf Stream is usually referred to the western boundary current north of Cape Hatteras (so north of the Southeast U.S. LME).**

We reworded this to say that the Southeast US LME "has a narrow shelf and is dominated by the western boundary current".

**3. Description of the forecast-observation mean bias (for Figs 5-7) could be more focused on those forecasts that have significant forecast-observation correlation coefficient (Figs 2-4).**

In the revised manuscript we now mention that the SST biases in the SS and NEUS LMEs are improved in the downscaled model for forecasts verifying in autumn and early winter when the downscaled forecasts have the most skill.

**4. Lines 278-284: authors could just write out the season name, instead of "first season", "last season", and "seasons 0 and 2".**

In the revised version we have replaced "seasons 0 and 2" with "the first and third seasons".

**5. Lines 294-295:**
**(1) "remaining three regions" -> "remaining four regions"?**

Yes, we have fixed this to now read "remaining four regions".

**(2) "aside from the increased spread in the Southeast U.S." not sure why it is "increased" when comparing with SS and NEUS based on Figure 9, please consider rephrasing this sentence.**

We agree that this was worded confusingly. It now reads "Differences between the two models are smaller in the remaining four regions, aside from NWA12 having higher spread than SPEAR in the Southeast U.S. and lower spread and RMSE than SPEAR in the Floridian region".

**6. Line 340: "mid-Atlantic Bight" -> MAB**

We appreciate this suggestion and have also replaced a few other instances of "Mid-Atlantic Bight" with MAB.

**7. Figure 11: Please indicate correlation significance in the figure for each panel. Figure 11 shows the correlation in the GOM is minimum at Lead 6 but increases at Lead 7. Could you please explain it? Please consider adding the location of the Scotian Shelf box in Figure 1.**
**-and-**
**8. Figure 12-13: please consider adding correlation significance in each correlation map.**

We have a semi transparent gray shading to the regions where the correlation is not significant at alpha = 0.1 in Figures 11, 12, and 13. A copy of these figures is included below. We have also added an outline of the Scotian Shelf box to the first panel of each figure.

**Nudged analysis simulation**

[Figure]

Correlation between Scotian Shelf SSS and simulated SSS in subsequent months

**NWA12 downscaled forecasts**

[Figure]

Correlation between Scotian Shelf SSS and simulated SSS in subsequent months

**SPEAR forecasts**

[Figure]

Correlation between Scotian Shelf SSS and simulated SSS in subsequent months

**9. Figure 20: Do predictions of bottom temperature also require approximately 4 ensemble members to provide a reasonable compromise between computational costs and prediction skill?**

Yes, bottom temperature and surface salinity have similar patterns of skill vs ensemble size. To show this, we have revised figure 20 to include panels for bottom temperature and surface salinity (figure also included below). We have also revised the text to mention that the effect of ensemble size is similar for all three variables.

[Figure]

**NEUS_LME skill vs. ensemble size**

(a) SST anomaly correlation
(b) Bottom temp. anomaly correlation
(c) Surface salinity correlation
(d) SST anomaly CRPS
(e) Bottom temp. anomaly CRPS
(f) Surface salinity CRPS

**Reviewer #3**

This article discusses the skill of historical seasonal forecasts utilizing a regional North Atlantic ocean model initialized with the GLORYS12 ocean analysis (approximately the same resolution as the ocean model) and forced by atmospheric forecast conditions from the SPEAR seasonal forecast. Results are then compared with the global, low resolution, coupled forecasts of SPEAR.

I have always been an advocate for the usage of ocean reanalysis as a tool for both initialization of forecasts – and for their use as a diagnostic tool to assess the ocean in regions where observations are sparse, or non-existent. However, the usage here, to use the GLORYS12 product as initialization for the regional seasonal forecasts – and then to assess skill against the GLORYS12 reanalysis seems somewhat incestuous to me – especially since the SPEAR ocean analysis likely differs substantially from the GLORYS12 analysis The study then really becomes one of assessing the initialization of a high resolution ocean model with GLORYS12 versus initializing with the ocean model component of SPEAR, and not particularly a "downscaling" of a seasonal forecast.

My question to the authors: If this system was to become an operational forecast system for the U.S. East Coast, would the goal be to initialize such forecasts with the real time

**GLORYS12 analysis (Skill assessed in this manuscript), or initialized by downscaling the SPEAR ocean analysis (Skill not assessed in this manuscript). If it is the former, then perhaps utilizing the full multi-model ensemble of NMME, as opposed to only SPEAR atmospheric component forcing, would be a more prudent approach, as that likely would increase the underdispersiveness of only using SPEAR atmospheric forecast forcing.**

**Recommendation: Despite my trepidation with regards to the skill assessment primarily against the system initialization product, I would recommend publication after the authors answer my question and comments.**

We appreciate the reviewer for raising these concerns with the methods and how the results are discussed. Ultimately, we believe that the methodology we employed is the best for examining the question at hand—whether a high resolution model can produce skillful seasonal predictions in the study region—and we plan to employ the same methodology (initializing with a simulation nudged towards the GLORYS12 reanalysis) for a quasi-operational product as part of NOAA's Climate, Ecosystems, and Fisheries Initiative. If the GLORYS reanalysis was a poor quality product with severe biases, we agree that using it for initialization and verification would give a biased assessment of prediction skill. However, if the GLORYS reanalysis was a perfect match for actual observations, aside from observation error, then using it for initialization and verification would be ideal. Since, as we cite in the manuscript, the GLORYS reanalysis has been found to match observations more closely than many other reanalysis, we argue that it is appropriate to use GLORYS for both initialization and verification.

To quantitatively explore this question, we compared the sea surface temperature forecasts from the MOM6-NWA12 and SPEAR models with the OISST v2 dataset instead of the GLORYS reanalysis. As we note in an answer below, the OISST dataset was assimilated by the SPEAR model, so this would potentially bias the skill assessment in favor of SPEAR. However, we see no meaningful difference whether OISST is used as the observations (figure below) or GLORYS is used as in the manuscript. In fact, at most lead times the skill of MOM6-NWA12 actually increases if OISST is used as the observations.

**Sea surface temperature correlation**

[Figure]

We also note that Jacox et al. 2023 ("Downscaled seasonal forecasts for the California Current System", PLOS Climate) similarly compared their seasonal forecasts with the same reanalysis

that was used for initialization, and they likewise found it made little difference if the skill assessment was performed using the GLORYS or OISST products instead.

In terms of whether or not this is truly a study of downscaling because the initial conditions are not derived from the model being downscaled, we note Jacox et al. 2023 and Kearney et al. 2021 also initialized from a different source yet still referred to their forecasts as "downscaled".

We have added two paragraphs to the Discussion in reply to this comment:

*The analysis showed that the high resolution regional model had significantly higher forecast skill than the global model in many cases, and that this skill comes from several sources including better representation of re-emergence, advection of water masses, and Gulf Stream variability and trends. Given the experimental design used in this study, however, where the high resolution model uses initial conditions from a different, higher-resolution source, it is difficult to determine how much of the increased forecast skill comes from the higher resolution initial conditions and how much comes from evolving the initial conditions forward in time with higher resolution. In an analysis for the U.S. West Coast, Jacox et al. (2023) examined two sets of downscaled retrospective forecasts, one initialized from a high resolution reanalysis (similar to the present study) and the other initialized from the coarse resolution parent model. Initializing from the high resolution reanalysis yielded generally negligible improvements in forecast skill for surface and bottom temperature, aside from in the first month. The high resolution reanalysis did significantly improve the skill of sea surface heights in their analysis. However, it is worth noting that improved (bias-corrected) atmospheric forcing was also included in their model runs initialized from the high resolution product. Overall, additional experiments are needed to conclusively determine the role of the resolution of the initial conditions in downscaled seasonal forecast skill.*

*As we noted in Section 2.4, the skill assessment could have been biased in favor of the high resolution model, which was initialized with the same GLORYS reanalysis used as the observations in the assessment. On the other hand, the GLORYS reanalysis has been repeatedly found to closely match in situ observations (Amaya et al., 2023; Carolina Castillo-Trujillo et al., 2023), which would suggest that comparing against the GLORYS reanalysis should be similar to comparing against in situ observations. To determine whether any bias could be an issue, we repeated the assessment of forecast SST anomaly correlation using the OISST dataset (Reynolds et al., 2007) instead of the GLORYS reanalysis (Figure S1). Even though the SPEAR model derived its initial conditions by assimilating data from OISST, there is no meaningful difference between the forecast skill relative to GLORYS or OISST. In fact, in many cases the downscaled model has slightly higher prediction skill if OISST is used as the observations. A lack of sensitivity to the dataset used for verification was also found by Jacox et al. (2023) who downscaled seasonal forecasts for the U.S. West Coast.*

**Itemized Comments:**

**1. I believe other studies (not seasonal forecasts, however) have been undertaken with 1/12th degree North Atlantic systems, although admittedly I could not find a particularly relevant study in my quick search. Perhaps the authors could [be] more explicit with regards to the definition of their 1/12th grid: Is the grid identical to a North Atlantic**

**subset of the GLORYS12 grid, or how does it differ from the ORCA12 grid utilized by GLORYS?**

We refer to Ross et al. 2023 at the beginning of the methods section for details about the model grid and configuration. To address the reviewer's comment, however, we have added a note in the revised version that the NWA12 model grid is a subset of the North and Equatorial Atlantic model grid from Chassignet and Xu (2017). The original model by Chassignet and coauthors is a HYCOM-based model, and the grid has no relation to the GLORYS/ORCA12 grid.

**2. It is not the responsibility of the authors to discuss the ocean initialization of SPEAR, but nonetheless, how it is initialized, and in particular, how its ocean state estimation approach differs from GLORYS12 is an important component of this study. More information is required to assess this, preferably with some explicit text in the manuscripts, but minimally by explicit citations of the SPEAR ocean initialization procedure. The manuscripts does show "0 lead" (actually 0.5 lead I believe) results that can be used to assess these differences somewhat, but some more explicit comparisons, particularly for the reemergence discussion would be useful – for instance, the manuscript shows the reemergence in the GLORYS12 reanalysis – is it also present in the SPEAR ocean analysis (or concatenation of 0 leads).**

We have added a sentence to the methods section: "Ocean initial conditions for the SPEAR retrospective forecasts were obtained by assimilating the OISSTv2 sea surface temperature product, vertical profiles from Argo floats, and several other sources using an Ensemble Adjustment Kalman Filter; see Lu et al. (2020) for details."

**3. The statement in the conclusion, " Finally, full data assimilation, rather than nudging towards a reanalysis, could improve prediction skill through better initial conditions · · ·," might be true – but not necessarily when basing that skill on the reanalysis being nudged towards. This may be particularly true if not many observations are going into the ocean analysis in the areas of skill assessment, which unfortunately may be true for the coastal region, strong ocean current (short ARGO float retention) regions under study in this manuscript.**

We believe we have addressed this comment with our reply to the beginning general comment.

**4. The spread error discussion was interesting, and the skill versus ensemble size (including CRPS results) did expand on this. But I am always interested in expanding on the probabilistic nature of the ensemble – and it would seem the reemergence diagnostics utilized here might be a natural way to expound on this. I assume the reemergence diagnostics are perform on the ensemble mean? Could an member by member diagnostics be performed that might lead to a "probability" of re-emergence that could be accessed for skill (Brier Score)?**

This is an interesting suggestion, but we believe that developing a probabilistic measure of re-emergence would be best suited as a topic for a future manuscript. The reviewer is correct that in the present manuscript we are using the ensemble mean for the reemergence diagnostic. We have not revised the text in response to this comment.

**5. I remind the authors that Atlantic Overturning Circulation variability can be driven by atmospheric variability as well (Jackson et al, 2016; https://doi.org/10.1038/ngeo2715), with particular implications to density anomalies along western Atlantic.**

We appreciate the reminder. We added a citation to this paper where we think it is most relevant: to support our remark that anomalies entering our model domain from the northern boundary typically take more than 1 year to reach the LMEs of interest in our study.

**6. The authors should highlight their skill assessment of ocean currents is performed using an independent "observation" source, and therefore is not as sensitive to the initial conditions as their temperature and salinity skill assessment. Although it then may be instructive to give some evidence of current skill in the GLORYS12 analysis (Aijaz et al, 2023; https://doi.org/10.1016/j.ocemod.2023.102241)**

Although technically the assessment of ocean currents is based on a satellite altimetry dataset rather than the GLORYS12 reanalysis, GLORYS assimilates the altimetry data and in practice the sea surface heights of the two datasets are similar. Furthermore, based on our comparison with the OISST dataset and the other sources we cited, we do not believe that in this study it makes a difference whether the dataset being used for evaluation is also used for initialization. We have not revised the text in response to this comment.

**7. In light of the previous two points, I wonder why authors did not present a more detailed skill assessment of currents beyond just a trend analysis?**

The present study is primarily focused on metrics that are readily translatable to fisheries management, such as surface and bottom temperature, and that also provide an overall picture of the model skill (for example, accurately predicting surface salinity requires accurately simulating advection and mixing). Ocean currents are not currently used by the marine resource managers we are working with. Comparison with the 1° global model is also potentially less interesting and relevant for ocean currents; for example, small coastal currents like the Eastern Maine Coastal Current *are* potentially relevant to fisheries and water quality, but they are obviously not represented in the 1° model. We believe an analysis of prediction skill for coastal currents in the high resolution model would be an interesting and useful subject for a separate study.

**8. I found the divergenging color schemes used in plots to not be particularly easy to distinguish null results, particulary with the red/green scheme used in figures 1-4 (plus I believe it is not particularly colour blind friendly). The purple/green scheme of figures 5-7**

**seems somewhat better – but either an explicitly white color marker for 0 difference, or 3 colour scheme (i.e. yellow as zero difference) might be preferable.**

We appreciate this suggestion to improve the accessibility of the paper. In the revised version, we have replaced the red/green color scheme with the same purple/green scheme used in figures 5–7. At the reviewer's suggestion, we also experimented with adding a pure white color to the middle of the colormap, but we did not think that this enhanced the readability of the figure. A sample revised figure is included below.

**Sea surface temperature correlation**

[Figure]

---

## Referee Report (RR1)

**Review of: Dynamically downscaled seasonal ocean forecasts for North American East Coast ecosystems**

by Andrew C. Ross, Charles A. Stock, Vimal Koul, Thomas L. Delworth, Feiyu Lu, Andrew Wittenberg

October 11, 2024

**Manuscript Synopsis**

2nd Review (although 1st review was done under tight time constraints and without access to usual sources of published research). This publication introduces a regional seasonal ocean forecast system for the United States east coast forced by the global SPEAR seasonal forecast and initialized with a subset of the GLORYS12 $\frac{1}{12}^{\circ}$ global ocean re-analysis over the domain in question.

I feel I still need to push back on the major point of my first review, which is the treatment of the GLORYS12 [Lellouche et al., 2013] re-analysis as both the verifying truth and initial conditions. As a producer of ocean analysis, I do not want to discourage their use for either the initialization of forecast, nor as a tool for verification. Verification of both atmospheric and ocean forecasts against their corresponding atmospheric or ocean analysis is done all the time – although I should add, not without its detractors, which I do not count myself as one – indeed, I regularly engage in the process. However, dynamical ocean and atmospheric analysis, as the authors are well aware – they spend several paragraphs of their introduction explaining why they wish to produce a dynamical forecast as opposed to existing statistical forecasts, which ultimately is the same achievement – is a best fit of the observations within a dynamically balanced system to obtain a best guess estimate of the ocean or atmosphere, at least on the grid and resolution in question. Uncertainty is **always** associated with this estimate, especially when observations are sparse, as they almost always are for the ocean sub-surface, or ocean surface salinity. The former is substantially improved by ARGO (effectively sometime around 2005), and the latter could be corrected by remotely sensed sea surface salinity as is currently provided by the SMOS European Space Agency satellite, and formerly by the NASA Aquarius mission – but this is not as yet a standard assimilation observation in ocean analysis like GLORYS12.

The authors have made some attempts to elicit this in the manuscript, but I think it is important (at least to me) that this gets further discussed in the manuscript. **However,** despite a rather long list of major comments, I am really only asking for a very minor change as suggested in item #6. My goal is to inform – not to impose major changes on the authors' manuscript. For this reason, and the fact I unfortunately either missed, or forgot to include some minor points in my original recommendation, my recommendation is still for some **Minor Revisions** prior to publication.

**My recommendation is Minor Revisions**

**Major Comments**

1. Thank you for addressing all my concerns, and other reviewers concerns, with the earlier version of the manuscript. The remaining items amongst the major comments are some further push back to these responses. I do not expect any major structural changes to the manuscript, and indeed in the end suggest only a minor change as requested in item #6. Most of the minor comments are unfortunately minor points I missed in doing the last review – some noted last time, but omitted from my review (access was lost to the marked-up pdf), or simply not noted previously.

2. As I stated in the synopsis: Although I do appreciate the authors attempts to recognize that the analysis do indeed come with some uncertainty, and I do highlight these below, I do believe some

further discussion, particularly when discussing the proposed skill of the downscaled system, is still warranted. Points where the authors have established this uncertainty are:

- ll. 127-128. We acknowledge, though, that deriving the initial conditions from a data assimilation process or a more sophisticated nudging method could improve the forecast spread and skill.

- ll. 277-280. It should be noted that the GLORYS12 reanalysis used as the observations in this comparison does not simulate salinity as well as it does temperature, and some of the reduction in bias may be due to the use of this reanalysis in the derivation of the initial conditions for the downscaled forecasts.

3. However, neither of the references they use to highlight the qualities of the GLORYS12 reanalysis [Amaya et al., 2023, Carolina Castillo-Trujillo et al., 2023], which are multi-system intercomparisons of which GLORYS12 is included, investigate whether the multi-model ensemble mean provides a better quality than GLORYS12. I am not recommending that the use of a multi-system ensemble mean would be a better set of initial condition – it would not be a valid dynamical state for one – but rather wish to again highlight there is uncertainty in the GLORYS12 analysis. The authors may wish to see Toyoda et al. [2015], which is a (now old) multi-system comparison of mixed layer depths for a set of global reanalysis, of which a much older version of GLORYS (0.25°) is contributing. Northern winter mixed layer depths (Figure 11 of the article) along the North American east coast are amongst the most uncertain (largest normalized spread), indicating any analysis in that region is likely to be uncertain – particularly before 2005, after which ARGO becomes well established, as shown in Figure 1 of Storto et al. [2019].

4. The example of validating against OISST SST given as a counter-example in the authors' response to reviewers comments is likely the least interesting for me. SST is by far the most observed variable in the ocean, with satellite remote sensing able to accurately observe the sea surface temperature to a nominal resolution of 4km. Cloud cover or rain may deplete those high resolution observations on a temporal basis, but I would expect different SST analysis to not actually differ substantially. So the fact that NWA12 performs better than SPEAR when validated with OISST SST analysis assimilated by the Kalman Filter assimilation in SPEAR is not really surprising to me.

5. Sea surface salinity (SSS) in not observed well in the ocean as already commented on by the authors. Similarly subsurface T/S profiles, mostly measured through ARGO floats that do not have long retention periods in the Gulf Stream / western boundary current areas of interest in this manuscript, are relatively sparsely observed. I would very strongly suspect that GLORYS12 will perform much better than the 1° SPEAR Kalman Filter assimilation in correctly estimating the subsurface, especially since the additional assimilation of sea surface height, in addition to GLORYS12 higher resolution giving it the ability to constrain the (at least larger scale) mesoscale activity that is known to improve ocean water mass properties in the analysis throughout the water column in the absence of any local InSitu observations [Fujii et al., 2024]. So the results I believe are really dependent on initial conditions, especially in the light of the fact that persistence is particularly skillful at predicting this, would be the bottom temperature results.

6. Ultimately, I believe I would be satisfied if one further statement (modified to the authors preference) is added to their discussion of ll. 233-235: Forecast correlation coefficients are higher for bottom temperature (Figure 3), which partially reflects its increased persistence. Most downscaled forecast correlations are higher than the persistence correlation, however, though the majority are not significantly higher. *This is perhaps not unexpected, as the downscaled NWA12 bottom temperatures are initialized by the exact persisted values coming from the GLORYS12 analysis used as validation. The lower correlated SPEAR bottom temperature values will likely be initialized somewhat differently.*

7. I would like to respond to the authors' response to the use of the AVISO gridded current product (DOI:10.48670/moi-00148). Yes, they are both Copernicus Marine Products. Yes, they are based on the same set of satellite data. No they **may** not be as similar as the authors state – the GLORYS12 currents being more than just geostrophic for one. I haven't done a comparison, and do not know of any comparison off hand – although I would be surprised if it is not in the Copericus quality

assessment document (QUID) of one, or both the products. It certainly was an omission to not include an independent observation only source of gridded currents in the Aijaz et al. [2023] manuscript, as this certainly was a standard in most of the ORA-IP papers [Toyoda et al., 2015, Shi et al., 2015, Palmer et al., 2015, Storto et al., 2015, Uotila et al., 2019]. This, however, harkens back to the previous point: There will be uncertainty in the assessment of currents, particularly near the coast, where the altimetry data is not as reliable (has larger error). The GLORYS12 product certainly was one of the better performing products in Aijaz et al. [2023], but not universally so, and likely the difference between the analysis is more indicative of this uncertainty, than it is of the individual analysis performance – in other words, the multi-system ensemble almost always out performs the member systems (again, not particularly explored in Aijaz et al. [2023]), but well explored in the other ORA-IP papers, even when mixing higher and lower resolution systems. Ultimately, the current assessment done in the manuscript here, which is only a trend assessment, does not depend too highly on this, and I will leave it there. No action required.

**Minor Comments**

1. ll. 39-44. The NAO was shown to be predictable in Scaife et al. [2014]. However, Smith et al. [2020] do explain why large ensemble would be necessary to pull the low NAO signal out of the noise in the application of the seasonal forecasts to real world scenarios, although this it really first developed in Eade et al. [2014]. That being said, the SPEAR journal publication [Delworth et al., 2020] does not give any details on whether any NAO predictiability is present in SPEAR, I suspect not as I don't believe they will get a significant signal with only a $1°$ ocean model (at least $\frac{1}{4}°$ is needed), but this is my own personal prejudice. Without the NAO signal available in the atmospheric forcing from the parent model, the regional model will not benefit from this possibility, no matter the number of ensemble members. If there are oceanic precursors for the NAO, the initializing GLORYS12 ocean analysis will almost certainly contain them, but they will not be properly integrated forward without the signal in the driving atmosphere. However, reemergence of existing ocean signatures could conceivably precede without too much undo intervention required by the atmosphere. The driving atmosphere is all beyond the authors' control – although it would be helpful if the authors could give some statement on SPEARS ability to forecast the NAO if that could be tracked down from existing analysis of the system.

2. ll. 68-72. I will just gently prod the authors that they could have investigated whether the 30 low resolution SPEAR members could have outperformed the 10 high resolution NWA12 members as an addition to the provided 10 to 10 member analysis of skill. But of course, that was not the purpose of the manuscript. Perhaps the authors' may with to say why they choose not to pursue this.

3. ll. 115,118. The manuscript does not actually provide a citation for GLORYS12 [Lellouche et al., 2013]?

4. ll. 153-155. I believe the authors are trying to find ways to improve their boundary conditions without breaking the seasonal hindcast paradigm of not using future information – which would prevent them from using GLORYS as they do in the Ross et al. [2023] reanalysis-forced historical simulation. A common way of using other than climatological boundary conditions in long term forecasts is to use an anomaly persistence, or damped anomaly persistence at the boundaries. In other words, use the GLORYS12 reanalysis as initial boundary conditions, but instead of continuing to use the GLORYS12 reanalysis, use the climatological boundary conditions plus the initial condition **anomaly** of GLORYS12 with respect to the climatology. You then potentially have the opportunity to damp the provided anomaly over some time period. An example of using a climatologically evolving persisted anomaly with the SST boundary condition of a (formerly) atmosphere only system can be found in Lin et al. [2016].

5. Figures 11, 12 & 13, SubSection 4.1.2. Firstly I apologize, I realize it is late in the game, and this is one comment I know I was going to comment on in my first review, but then failed to do so, but it is really hard to follow the discussion of Section 4.1.2 by having to flip back and forth between the text and at least 2 of 3 figures – even with modern ways of doing things, like having 4 versions of

the manuscript lined up across my two screens. I think you would do your readers a favour – and significantly help the weight of your argument if one could have all three results (i.e. Figures 11, 12 & 13 ) lined up as a single figure. For instance, if you restricted yourself to a snapshot every 2nd month (starting at 0), that would likely be frequent enough to show the propagation in each system, and then one could have the 3 systems (analysis, forecast, SPEAR forecast) lined up vertically, with the 6 time snapshots arranged horizontally. There is ample white space to the left and right to do this with no loss of magnification. But if you did insist on keeping the current 12 snapshots, one could still consider having a full page figure with the 3 systems across and the 12 snapshots vertically, but this would likely result in a decrease in the individual snapshot sizes. Please consider doing one of these two options.

6. The authors may wish to know – and the reader may wish to know too, that there are theoretical formula that can explain the dependence on number of ensemble members in the presence of ensemble spread versus error for the CRPS in equation 3 of Leutbecher [2019], or the number of ensemble members in the presence of average correlation between members and average correlation with observation of the individual members in the correlation of the ensemble mean with observation with equation 2 of Murphy [1990], p. 99. Indeed, for CRPS, one can define a "fair CRPS" score that would be independent of ensemble size as in equation 4 of Leutbecher [2019].

7. ll. 525-528. My impression from the bias shown in Figures 5-7 is that they may be caused by errors in (too much) vertical mixing, and are not necessarily linked to the external/coupled atmospheric forcing. This impression is due to the phase relation ($\sim 90°$) between surface and bottom – and could of course be an incorrect impression. Correcting the fluxes may not be the correction required.

**References**

Saima Aijaz, Gary B. Brassington, Prasanth Divakaran, Charly Régnier, Marie Drévillon, Jan Maksymczuk, and K. Andrew Peterson. Verification and intercomparison of global ocean eulerian near-surface currents. *Ocean Modelling*, 186:102241, 2023. ISSN 1463-5003. doi: https://doi.org/10.1016/j.ocemod.2023.102241. URL https://www.sciencedirect.com/science/article/pii/S1463500323000823.

Dillon J. Amaya, Michael A. Alexander, James D. Scott, and Michael G. Jacox. An evaluation of high-resolution ocean reanalyses in the california current system. *Progress in Oceanography*, 210:102951, 2023. ISSN 0079-6611. doi: https://doi.org/10.1016/j.pocean.2022.102951. URL https://www.sciencedirect.com/science/article/pii/S0079661122002105.

Alma Carolina Castillo-Trujillo, Young-Oh Kwon, Paula Fratantoni, Ke Chen, Hyodae Seo, Michael A. Alexander, and Vincent S. Saba. An evaluation of eight global ocean reanalyses for the northeast u.s. continental shelf. *Progress in Oceanography*, 219:103126, 2023. ISSN 0079-6611. doi: https://doi.org/10.1016/j.pocean.2023.103126. URL https://www.sciencedirect.com/science/article/pii/S0079661123001696.

Thomas L. Delworth, William F. Cooke, Alistair Adcroft, Mitchell Bushuk, Jan-Huey Chen, Krista A. Dunne, Paul Ginoux, Richard Gudgel, Robert W. Hallberg, Lucas Harris, Matthew J. Harrison, Nathaniel Johnson, Sarah B. Kapnick, Shian-Jian Lin, Feiyu Lu, Sergey Malyshev, Paul C. Milly, Hiroyuki Murakami, Vaishali Naik, Salvatore Pascale, David Paynter, Anthony Rosati, M.D. Schwarzkopf, Elena Shevliakova, Seth Underwood, Andrew T. Wittenberg, Baoqiang Xiang, Xiaosong Yang, Fanrong Zeng, Honghai Zhang, Liping Zhang, and Ming Zhao. Spear: The next generation gfdl modeling system for seasonal to multi-decadal prediction and projection. *Journal of Advances in Modeling Earth Systems*, 12(3):e2019MS001895, 2020. doi: https://doi.org/10.1029/2019MS001895. URL https://agupubs.onlinelibrary.wiley.com/doi/abs/10.1029/2019MS001895. e2019MS001895 2019MS001895.

Rosie Eade, Doug Smith, Adam Scaife, Emily Wallace, Nick Dunstone, Leon Hermanson, and Niall Robinson. Do seasonal-to-decadal climate predictions underestimate the predictability of the real world? *Geophysical Research Letters*, 41(15):5620–5628, 2014. ISSN 1944-8007. doi: 10.1002/2014GL061146. URL http://dx.doi.org/10.1002/2014GL061146. 2014GL061146.

Yosuke Fujii, Elisabeth Remy, Magdalena Alonso Balmaseda, Shoichiro Kido, Jennifer Waters, K Andrew Peterson, Gregory Christopher Smith, Ichiro Ishikawa, and Kamel Chikhar. The international multi-system oses/osses by the un ocean decade project synobs and its early results. *Frontiers in Marine Science*, 11, 2024. doi: 10.3389/fmars.2024.1476131.

J.-M. Lellouche, O. Le Galloudec, M. Drévillon, C. Régnier, E. Greiner, G. Garric, N. Ferry, C. Desportes, C.-E. Testut, C. Bricaud, R. Bourdallé-Badie, B. Tranchant, M. Benkiran, Y. Drillet, A. Daudin, and C. De Nicola. Evaluation of global monitoring and forecasting systems at mercator océan. *Ocean Science*, 9(1):57–81, 2013. doi: 10.5194/os-9-57-2013. URL `https://os.copernicus.org/articles/9/57/2013/`.

Martin Leutbecher. Ensemble size: How suboptimal is less than infinity? *Quarterly Journal of the Royal Meteorological Society*, 145(S1):107–128, 2019. doi: https://doi.org/10.1002/qj.3387. URL `https://rmets.onlinelibrary.wiley.com/doi/abs/10.1002/qj.3387`.

Hai Lin, Normand Gagnon, Stephane Beauregard, Ryan Muncaster, Marko Markovic, Bertrand Denis, and Martin Charron. Geps-based monthly prediction at the canadian meteorological centre. *Monthly Weather Review*, 144(12):4867 – 4883, 2016. doi: 10.1175/MWR-D-16-0138.1. URL `https://journals.ametsoc.org/view/journals/mwre/144/12/mwr-d-16-0138.1.xml`.

J. M. Murphy. Assessment of the practical utility of extended range ensemble forecasts. *Quarterly Journal of the Royal Meteorological Society*, 116(491):89–125, 1990. doi: https://doi.org/10.1002/qj.49711649105. URL `https://rmets.onlinelibrary.wiley.com/doi/abs/10.1002/qj.49711649105`.

M.D. Palmer, C.D. Roberts, M. Balmaseda, Y.-S. Chang, G. Chepurin, N. Ferry, Y. Fujii, S.A. Good, S. Guinehut, K. Haines, F. Hernandez, A. Köhl, T. Lee, M.J. Martin, S. Masina, S. Masuda, K.A. Peterson, A. Storto, T. Toyoda, M. Valdivieso, G. Vernieres, O. Wang, and Y. Xue. Ocean heat content variability and change in an ensemble of ocean reanalyses. *Climate Dynamics*, pages 1–22, 2015. ISSN 0930-7575. doi: 10.1007/s00382-015-2801-0. URL `http://dx.doi.org/10.1007/s00382-015-2801-0`.

A. C. Ross, C. A. Stock, A. Adcroft, E. Curchitser, R. Hallberg, M. J. Harrison, K. Hedstrom, N. Zadeh, M. Alexander, W. Chen, E. J. Drenkard, H. du Pontavice, R. Dussin, F. Gomez, J. G. John, D. Kang, D. Lavoie, L. Resplandy, A. Roobaert, V. Saba, S.-I. Shin, S. Siedlecki, and J. Simkins. A high-resolution physical–biogeochemical model for marine resource applications in the northwest atlantic (mom6-cobalt-nwa12 v1.0). *Geoscientific Model Development*, 16(23):6943–6985, 2023. doi: 10.5194/gmd-16-6943-2023. URL `https://gmd.copernicus.org/articles/16/6943/2023/`.

A. A. Scaife, A. Arribas, E. Blockley, A. Brookshaw, R. T. Clark, N. Dunstone, R. Eade, D. Fereday, C. K. Folland, M. Gordon, L. Hermanson, J. R. Knight, D. J. Lea, C. MacLachlan, A. Maidens, M. Martin, A. K. Peterson, D. Smith, M. Vellinga, E. Wallace, J. Waters, and A. Williams. Skillful long-range prediction of European and North American winters. *Geophysical Research Letters*, 41(7):2514–2519, 2014. ISSN 1944-8007. doi: 10.1002/2014GL059637. URL `http://dx.doi.org/10.1002/2014GL059637`. 2014GL059637.

L. Shi, O. Alves, R. Wedd, M.A. Balmaseda, Y. Chang, G. Chepurin, N. Ferry, Y. Fujii, F. Gaillard, S.A. Good, S. Guinehut, K. Haines, F. Hernandez, T. Lee, M. Palmer, K.A. Peterson, S. Masuda, A. Storto, T. Toyoda, M. Valdivieso, G. Vernieres, X. Wang, and Y. Yin. An assessment of upper ocean salinity content from the ocean reanalyses inter-comparison project (ora-ip). *Climate Dynamics*, pages 1–21, 2015. ISSN 0930-7575. doi: 10.1007/s00382-015-2868-7. URL `http://dx.doi.org/10.1007/s00382-015-2868-7`.

D. M. Smith, A. A. Scaife, R. Eade, P. Athanasiadis, A. Bellucci, I. Bethke, R. Bilbao, L. F. Borchert, F. Caron, L.-P.and Counillon, G. Danabasoglu, T. Delworth, F. J. Doblas-Reyes, N. J. Dunstone, S. Estella-Perez, V.and Flavoni, L. Hermanson, N. Keenlyside, V. Kharin, M. Kimoto, W. J. Merryfield, J. Mignot, T. Mochizuki, K. Modali, W. A. Monerie, P.-A.and Müller, D. Nicolí, P. Ortega, K. Pankatz, H. Pohlmann, J. Robson, P. Ruggieri, R. Sospedra-Alfonso, D. Swingedouw, Y. Wang, S. Wild, S. Yeager, X. Yang, and L. Zhang. North atlantic climate far more predictable than models imply. *Nature*, 583: 796–800, 2020. doi: 10.1038/s41586-020-2525-0. URL `https://doi.org/10.1038/s41586-020-2525-0`.

Andrea Storto, Simona Masina, Magdalena Balmaseda, Stéphanie Guinehut, Yan Xue, Tanguy Szekely, Ichiro Fukumori, Gael Forget, You-Soon Chang, SimonA. Good, Armin Köhl, Guillaume Vernieres, Nicolas Ferry, K.Andrew Peterson, David Behringer, Masayoshi Ishii, Shuhei Masuda, Yosuke Fujii, Takahiro Toyoda, Yonghong Yin, Maria Valdivieso, Bernard Barnier, Tim Boyer, Tony Lee, Jérome Gourrion, Ou Wang, Patrick Heimback, Anthony Rosati, Robin Kovach, Fabrice Hernandez, MatthewJ. Martin, Masafumi Kamachi, Tsurane Kuragano, Kristian Mogensen, Oscar Alves, Keith Haines, and Xiaochun Wang. Steric sea level variability (1993–2010) in an ensemble of ocean reanalyses and objective analyses. *Climate Dynamics*, pages 1–21, 2015. ISSN 0930-7575. doi: 10.1007/s00382-015-2554-9. URL `http://dx.doi.org/10.1007/s00382-015-2554-9`.

Andrea Storto, Simona Masina, Simona Simoncelli, Doroteaciro Iovino, Andrea Cipollone, Marie Drevillon, Yann Drillet, Karina von Schuckman, Laurent Parent, Gilles Garric, Eric Greiner, Charles Desportes, Hao Zuo, Magdalena A. Balmaseda, and K. Andrew Peterson. The added value of the multi-system spread information for ocean heat content and steric sea level investigations in the CMEMS GREP ensemble reanalysis product. *Climate Dynamics*, 53(1-2):287–312, 2019. doi: 10.1007/s00382-018-4585-5. URL `https://app.dimensions.ai/details/publication/pub.1110638636`.

Takahiro Toyoda, Yosuke Fujii, Tsurane Kuragano, Masafumi Kamachi, Yoichi Ishikawa, Shuhei Masuda, Kanako Sato, Toshiyuki Awaji, Fabrice Hernandez, Nicolas Ferry, Stéphanie Guinehut, MatthewJ. Martin, K.Andrew Peterson, SimonA. Good, Maria Valdivieso, Keith Haines, Andrea Storto, Simona Masina, Armin Köhl, Hao Zuo, Magdalena Balmaseda, Yonghong Yin, Li Shi, Oscar Alves, Gregory Smith, You-Soon Chang, Guillaume Vernieres, Xiaochun Wang, Gael Forget, Patrick Heimbach, Ou Wang, Ichiro Fukumori, and Tong Lee. Intercomparison and validation of the mixed layer depth fields of global ocean syntheses. *Climate Dynamics*, pages 1–21, 2015. ISSN 0930-7575. doi: 10.1007/s00382-015-2637-7. URL `http://dx.doi.org/`.

Petteri Uotila, Hugues Goosse, Keith Haines, Matthieu Chevallier, Antoine Barth/'elemy, Clément Bricaud, Jim Carton, Neven Fučkar, Gilles Garric, Doroteaciro Iovino, Frank Kauker, Meri Korhonen, Vidar S. Lien, Marika Marnela, François Massonnet, Davi Mignac, K. Andrew Peterson, Remon Sadikni, Li Shi, Steffen Tietsche, Takahiro Toyoda, Jiping Xie, and Zhaoru Zhang. An assessment of ten ocean reanalyses in the polar regions. *Climate Dynamics*, 52:1613–1650, 2019. doi: 10.1007/s00382-018-4242-z. URL `https://doi.org/10.1007/s00382-018-4242-z`.

---

## Author Response (AR2)

**Reviewer #2**

**I am satisfied with the revisions made by the authors, and have only one last comment on the "nudging" method that authors used to the MOM6-NWA model:**

**In the methodology (lines116-118), authors suggest that temperature and salinity from the MOM6-NWA12 model "were nudged towards monthly means from the GLORYS12 reanalysis with a 90 day damping time scale", and "addition of nudging helps maximize the accuracy of the initial conditions" (lines 119-120). However, it is not described at all how the nudging is performed, e.g. the details of the nudging method and references. We only know it is not that "sophisticated" (line 128).**

We thank the reviewer for reviewing the manuscript again. In response to this comment, we have changed the text as follows (new text in italics):

[...] temperature and salinity throughout the model domain were nudged *(i.e., restored using Newtonian relaxation)* towards monthly means from the GLORYS12 reanalysis with a 90 day damping time scale.

This should ensure the reader understands that the nudging we used is the commonly known Newtonian relaxation.

**Reviewer #3**

**Manuscript Synopsis 2nd Review (although 1st review was done under tight time constraints and without access to usual sources of published research). This publication introduces a regional seasonal ocean forecast system for the United States east coast forced by the global SPEAR seasonal forecast and initialized with a subset of the GLORYS12 1 12◦ global ocean re-analysis over the domain in question.**

**I feel I still need to push back on the major point of my first review, which is the treatment of the GLORYS12 [Lellouche et al., 2013] re-analysis as both the verifying truth and initial conditions. As a producer of ocean analysis, I do not want to discourage their use for either the initialization of forecast, nor as a tool for verification. Verification of both atmospheric and ocean forecasts against their corresponding atmospheric or ocean analysis is done all the time – although I should add, not without its detractors, which I do not count myself as one – indeed, I regularly engage in the process. However, dynamical ocean and atmospheric analysis, as the authors are well aware – they spend several paragraphs of their introduction explaining why they wish to produce a dynamical forecast as opposed to existing statistical forecasts, which ultimately is the same achievement – is a best fit of the observations within a dynamically balanced**

**system to obtain a best guess estimate of the ocean or atmosphere, at least on the grid and resolution in question. Uncertainty is always associated with this estimate, especially when observations are sparse, as they almost always are for the ocean sub-surface, or ocean surface salinity. The former is substantially improved by ARGO (effectively sometime around 2005), and the latter could be corrected by remotely sensed sea surface salinity as is currently provided by the SMOS European Space Agency satellite, and formerly by the NASA Aquarius mission – but this is not as yet a standard assimilation observation in ocean analysis like GLORYS12.**

**The authors have made some attempts to elicite this in the manuscript, but I think it is important (at least to me) that this gets further discussed in the manuscript. However, despite a rather long list of major comments, I am really only asking for a very minor change as suggested in item #6. My goal is to inform – not to impose major changes on the authors' manuscript. For this reason, and the fact I unfortunately either missed, or forgot to include some minor points in my original recommendation, my recommendation is still for some Minor Revisions prior to publication.**

**1. Thank you for addressing all my concerns, and other reviewers concerns, with the earlier version of the manuscript. The remaining items amongst the major comments are some further push back to these responses. I do not expect any major structural changes to the manuscript, and indeed in the end suggest only a minor change as requested in item #6. Most of the minor comments are unfortunately minor points I missed in doing the last review – some noted last time, but omitted from my review (access was lost to the marked-up pdf), or simply not noted previously.**

We appreciate the reviewer's thoroughness and helpful comments. We have made the change requested in item #6, as well as in several other comments where a revision was suggested.

**2. As I stated in the synopsis: Although I do appreciate the authors attempts to recognize that the analysis do indeed come with some uncertainty, and I do highlight these below, I do believe some further discussion, particularly when discussing the proposed skill of the downscaled system, is still warranted. Points where the authors have established this uncertainty are:**
**• ll. 127-128. We acknowledge, though, that deriving the initial conditions from a data assimilation process or a more sophisticated nudging method could improve the forecast spread and skill.**
**• ll. 277-280. It should be noted that the GLORYS12 reanalysis used as the observations in this comparison does not simulate salinity as well as it does temperature, and some of the reduction in bias may be due to the use of this reanalysis in the derivation of the initial conditions for the downscaled forecasts.**

In response to comments #2, 3, and 6, we have made the revision ultimately requested in item #6.

**3. However, neither of the references they use to highlight the qualities of the GLORYS12 reanalysis [Amaya et al., 2023, Carolina Castillo-Trujillo et al., 2023], which are multi-system intercomparisons of which GLORYS12 is included, investigate whether the multi-model ensemble mean provides a better quality than GLORYS12. I am not recommending that the use of a multi-system ensemble mean would be a better set of initial condition – it would not be a valid dynamical state for one – but rather wish to again highlight there is uncertainty in the GLORYS12 analysis. The authors may wish to see Toyoda et al. [2015], which is a (now old) multi-system comparison of mixed layer depths for a set of global reanalysis, of which a much older version of GLORYS (0.25◦ ) is contributing. Northern winter mixed layer depths (Figure 11 of the article) along the North American east coast are amongst the most uncertain (largest normalized spread), indicating any analysis in that region is likely to be uncertain – particularly before 2005, after which ARGO becomes well established, as shown in Figure 1 of Storto et al. [2019].**

See above.

**4. The example of validating against OISST SST given as a counter-example in the authors' response to reviewers comments is likely the least interesting for me. SST is by far the most observed variable in the ocean, with satellite remote sensing able to accurately observe the sea surface temperature to a nominal resolution of 4km. Clould cover or rain may deplete those high resolution observations on a temporal basis, but I would expect different SST analysis to not actually differ substantially. So the fact that NWA12 performs better than SPEAR when validated with OISST SST analysis assimilated by the Kalman Filter assimilation in SPEAR is not really surprising to me. 5. Sea surface salinity (SSS) in not observed well in the ocean as already commented on by the authors. Similarly subsurface T/S profiles, mostly measured through ARGO floats that do not have long retention periods in the Gulf Stream / western boundary current areas of interest in this manuscript, are relatively sparsely observed. I would very strongly suspect that GLORYS12 will perform much better than the 1◦ SPEAR Kalman Filter assimilation in correctly estimating the subsurface, especially since the additional assimilation of sea surface height, in addition to GLORYS12 higher resolution giving it the ability to constrain the (at least larger scale) mesoscale activity that is known to improve ocean water mass properties in the analysis throughout the water column in the absence of any local InSitu observations [Fujii et al., 2024]. So the results I believe are really dependent on initial conditions, especially in the light of the fact that persistence is particularly skillful at predicting this, would be the bottom temperature results.**

We agree that one should not expect SST to show the largest differences between sources of data used for initialization or verification. We thought it presented a useful contrast, though, since data from GLORYS was used in the initialization of the downscaled model, whereas data from OISST was used in the initialization of the SPEAR model. These two datasets are also produced differently, with GLORYS being a full data assimilative analysis and OISST primarily deriving data from satellites. We have not changed the manuscript in response to this comment.

**6. Ultimately, I believe I would be satisfied if one further statement (modified to the authors preference) is added to their discussion of ll. 233-235: Forecast correlation coefficients are higher for bottom temperature (Figure 3), which partially reflects its increased persistence. Most downscaled forecast correlations are higher than the persistence correlation, however, though the majority are not significantly higher. This is perhaps not unexpected, as the downscaled NWA12 bottom temperatures are initialized by the exact persisted values coming from the GLORYS12 analysis used as validation. The lower correlated SPEAR bottom temperature values will likely be initialized somewhat differently.**

We appreciate the reviewer's suggestion and have revised the manuscript accordingly (new text in italics below):

In the Northeast U.S. (Figure 3d--f), the pattern of forecast skill and downscaling improvement is similar to that seen for surface temperature. In other regions, NWA12 bottom temperature predictions have significantly higher skill than SPEAR for some cases where they did not for surface temperature. *This is perhaps not unexpected, as the downscaled NWA12 bottom temperature forecasts were initialized with data from the same GLORYS12 reanalysis used as validation, and the lower correlated SPEAR bottom temperature values partially reflect the persistence of differences between the SPEAR initialization and the GLORYS12 reanalysis. With this caution in mind,* skill and improvement on the Scotian Shelf [...]

**7. I would like to respond to the authors' response to the use of the AVISO gridded current product (DOI:10.48670/moi-00148). Yes, they are both Copernicus Marine Products. Yes, they are based on the same set of satellite data. No they may not be as similar as the authors state – the GLORYS12 currents being more than just geostrophic for one. I haven't done a comparison, and do not know of any comparison off hand – although I would be surprised if it is not in the Copericus quality 2 assessment document (QUID) of one, or both the products. It certainly was an omission to not include an independent observation only source of gridded currents in the Aijaz et al. [2023] manuscript, as this certainly was a standard in most of the ORA-IP papers [Toyoda et al., 2015, Shi et al., 2015, Palmer et al., 2015, Storto et al., 2015, Uotila et al., 2019]. This, however, harkens back to the previous point: There will be uncertainty in the assessment of currents, particularly near the coast, where the altimetry data is not as reliable (has larger error). The GLORYS12 product certainly was one of the better performing products in Aijaz et al. [2023], but not universally so, and likely the difference between the analysis is more indicative of this uncertainty, than it is of the individual analysis performance – in other words, the multi-system ensemble almost always out performs the member systems (again, not particularly explored in Aijaz et al. [2023]), but well explored in the other ORA-IP papers, even when mixing higher and lower resolution systems. Ultimately, the current assessment done in the manuscript here, which is only a trend assessment, does not depend too highly on this, and I will leave it there. No action required.**

We thank the reviewer for sharing their knowledge on this topic.

**Minor Comments**

**1. ll. 39-44. The NAO was shown to be predictable in Scaife et al. [2014]. However, Smith et al. [2020] do explain why large ensemble would be necessary to pull the low NAO signal out of the noise in the application of the seasonal forecasts to real world scenarios, although this it really first developed in Eade et al. [2014]. That being said, the SPEAR journal publication [Delworth et al., 2020] does not give any details on whether any NAO predictiability is present in SPEAR, I suspect not as I don't believe they will get a significant signal with only a 1° ocean model (at least 1 4° is needed), but this is my own personal prejudice. Without the NAO signal available in the atmospheric forcing from the parent model, the regional model will not benefit from this possibility, no matter the number of ensemble members. If there are oceanic precursors for the NAO, the initializing GLORYS12 ocean analysis will almost certainly contain them, but they will not be properly integrated forward without the signal in the driving atmosphere. However, reemergence of existing ocean signatures could conceivably precede without too much undo intervention required by the atmosphere. The driving atmosphere is all beyond the authors' control – although it would be helpful if the authors could give some statement on SPEARS ability to forecast the NAO if that could be tracked down from existing analysis of the system.**

We believe this topic is best reserved for a different manuscript.

**2. ll. 68-72. I will just gently prod the authors that they could have investigated whether the 30 low resolution SPEAR members could have outperformed the 10 high resolution NWA12 members as an addition to the provided 10 to 10 member analysis of skill. But of course, that was not the purpose of the manuscript. Perhaps the authors' may with to say why they choose not to pursue this.**

We agree that this is beyond the scope of the present manuscript.

**3. ll. 115,118. The manuscript does not actually provide a citation for GLORYS12 [Lellouche et al., 2013]?**

We thank the reviewer for catching this oversight. In the newly revised version, we have added citations for the GLORYS12, ERA5, and GloFAS reanalyses where they are first mentioned.

**4. ll. 153-155. I believe the authors are trying to find ways to improve their boundary conditions without breaking the seasonal hindcast paradigm of not using future information – which would prevent them from using GLORYS as they do in the Ross et al. [2023] reanalysis-forced historical simulation. A common way of using other than climatological boundary conditions in long term forecasts is to use an anomaly persistence, or damped anomaly persistence at the boundaries. In other words, use the GLORYS12 reanalysis as initial boundary conditions, but instead of continuing to use the**

**GLORYS12 reanalysis, use the climatological boundary conditions plus the initial condition anomaly of GLORYS12 with respect to the climatology. You then potentially have the opportunity to damp the provided anomaly over some time period. An example of using a climatologically evolving persisted anomaly with the SST boundary condition of a (formerly) atmosphere only system can be found in Lin et al. [2016].**

Yes, we were careful to not include future information in the retrospective simulations. Our argument here was that due to the large extent of the regional model domain, the timescale for anomalies to propagate from the boundaries to the interior is longer than the 1 year length of the forecasts, so it does not matter much what we use for the open boundary condition data, and we can simplify things by only using climatology. Using persistence of damped persistence of anomalies is an interesting idea, though, that we will experiment with in the future. We have not changed the current version of the manuscript in response to this comment.

**5. Figures 11, 12 & 13, SubSection 4.1.2. Firstly I apologize, I realize it is late in the game, and this is one comment I know I was going to comment on in my first review, but then failed to do so, but it is really hard to follow the discussion of Section 4.1.2 by having to flip back and forth between the text and at least 2 of 3 figures – even with modern ways of doing things, like having 4 versions of 3 the manuscript lined up across my two screens. I think you would do your readers a favour – and significantly help the weight of your argument if one could have all three results (i.e. Figures 11, 12 & 13 ) lined up as a single figure. For instance, if you restricted yourself to a snapshot every 2nd month (starting at 0), that would likely be frequent enough to show the propagation in each system, and then one could have the 3 systems (analysis, forecast, SPEAR forecast) lined up vertically, with the 6 time snapshots arranged horizontally. There is ample white space to the left and right to do this with no loss of magnification. But if you did insist on keeping the current 12 snapshots, one could still consider having a full page figure with the 3 systems across and the 12 snapshots vertically, but this would likely result in a decrease in the individual snapshot sizes. Please consider doing one of these two options.**

We appreciate this suggestion for improving the readability of the manuscript. In the revised version, we have replaced the original 3 figures with a new single figure that shows all three datasets together. To keep the figure at a readable size, the new figure only shows 1, 3, 5, and 7 month lead times; this is enough to show the features mentioned in the text, and includes the 3 and 7 month lead times that are explicitly called out.

**6. The authors may wish to know – and the reader may wish to know too, that there are theoretical formula that can explain the dependence on number of ensemble members in the presence of ensemble spread versus error for the CRPS in equation 3 of Leutbecher [2019], or the number of ensemble members in the presence of average correlation between members and average correlation with observation of the individual members in the correlation of the ensemble mean with observation with equation 2 of Murphy [1990],**

**p. 99. Indeed, for CRPS, one can define a "fair CRPS" score that would be independent of ensemble size as in equation 4 of Leutbecher [2019].**

In response to this suggestion, we revised the manuscript to read (new text in italics):

[...] the improvement from a single member to two members is substantial, while the improvement from 4 to 10 members is minor. *This is consistent with the expected effect of ensemble size on estimation of the mean and CRPS (Leutbecher, 2018)*, [...]

**7. ll. 525-528. My impression from the bias shown in Figures 5-7 is that they may be caused by errors in (too much) vertical mixing, and are not necessarily linked to the external/coupled atmospheric forcing. This impression is due to the phase relation (~ 90∘ ) between surface and bottom – and could of course be an incorrect impression. Correcting the fluxes may not be the correction required.**

This is a fair point. We added a sentence to this paragraph:
*Reducing biases in ocean mixing, whether by correcting the wind forcing or adjusting the model parameterizations, may be especially important for reducing bottom temperature biases.*